# Axonal transport of CHMP2b is regulated by kinesin-binding protein and disrupted by CHMP2b[intron5]

Konner R Kirwan[1], Veria Puerta-Alvarado[1] , Clarissa L Waites[2,3]

**CHMP2b is a core component of the ESCRT pathway that catalyzes formation of multivesicular bodies for endolysosomal protein degradation. Although mutation/loss-of-function of CHMP2b promotes presynaptic dysfunction and degeneration, indicating its critical role in presynaptic protein homeostasis, the mechanisms responsible for CHMP2b localization and recruitment to synapses remain unclear. Here, we characterize CHMP2b axonal trafficking and show that its transport and recruitment to presynaptic boutons, as well as its cotransport with other ESCRT proteins, are regulated by neuronal activity. In contrast, the frontotemporal dementia–causative CHMP2b[intron5] mutation exhibits little processive movement or presynaptic localization in the presence or absence of neuronal activity. Instead, CHMP2b[intron5] transport vesicles exhibit oscillatory behavior reminiscent of a tug-of-war between kinesin and dynein motor proteins. We show that this phenotype is caused by deficient binding of CHMP2b[intron5] to kinesin-binding protein, which we identify as a key regulator of CHMP2b transport. These findings shed light on the mechanisms of CHMP2b axonal trafficking and synaptic localization, and their disruption by CHMP2b[intron5].**

## Introduction

Maintaining the neuronal proteome is essential for nervous system health, as evidenced by the many studies showing that disruption of cellular pathways responsible for protein synthesis, trafficking, and degradation can precipitate synaptic dysfunction and neurodegeneration (Esposito et al, 2012; Bezprozvanny & Hiesinger, 2013; Hall et al, 2017; Berth & Lloyd, 2023; Wilson et al, 2023). One such pathway, which mediates endolysosomal protein degradation, is the endosomal sorting complex required for transport (ESCRT). The ESCRT pathway comprises five heteromeric complexes (ESCRT-0, ESCRT-I, ESCRT-II, ESCRT-III, and Vps4) that sequentially assemble on endosomal membranes and function to sequester and sort proteins into intraluminal vesicles (ILVs), generating multivesicular bodies (MVBs) that deliver their cargo to lysosomes. ESCRT-0,

ESCRT-I, and ESCRT-II complexes play pivotal roles in cargo recognition and clustering, whereas ESCRT-III and Vps4 complexes mediate membrane deformation and scission, driving ILV formation (Henne et al, 2011). In neurons, the ESCRT pathway mediates protein degradation not only through its role in autophagosome closure (Takahashi et al, 2018), but also through the direct capture and sorting of proteins into MVBs (Sadoul et al, 2018). Although substrates of the ESCRT pathway were traditionally thought to be restricted to transmembrane proteins, our group and others have recently shown that they also include cytosolic proteins such as tau and α-synuclein (Vaz-Silva et al, 2018; Chen et al, 2019; Benyair et al, 2023; Nim et al, 2023), revealing the critical role of this pathway in clearing aggregation-prone and disease-relevant proteins.

This essential role is reflected by multiple studies showing that mutation or loss-of-function of ESCRT components induces neurodegeneration in mouse models (Lee et al, 2007, 2019; Tamai et al, 2008) and humans (Skibinski et al, 2005; Parkinson et al, 2006; Cox et al, 2010; Ghanim et al, 2010; Gao et al, 2017; Waegaert et al, 2022). In particular, mutations in the ESCRT-III protein charged multivesicular body protein 2b (CHMP2b), responsible for ILV scission through its recruitment of the membrane-severing AAA-ATPase Vps4 (Teis et al, 2008), result in frontotemporal dementia (FTD) and amyotrophic lateral sclerosis (Skibinski et al, 2005; Parkinson et al, 2006; Cox et al, 2010; Ghanim et al, 2010; Gao et al, 2017; Waegaert et al, 2022). The best characterized of these is CHMP2b[intron5], an autosomal dominant mutation discovered in a Danish pedigree of FTD-3 (Skibinski et al, 2005), in which a missense mutation occurring in the splice acceptor site of exon 6 replaces the final 36 amino acids with a single valine residue, causing a C-terminal truncation (Ugbode & West, 2021). Within this truncated region is a microtubule-interacting motif domain that serves as a binding site for Vps4 (Stuchell-Brereton et al, 2007; Ugbode & West, 2021). Consequently, CHMP2b[intron5] is unable to recruit Vps4 in order to initiate ILV scission, leading to impaired maturation of endosomes into functional MVBs (van der Zee et al, 2008; Urwin et al, 2010). However, the loss of a microtubule-interacting motif domain and surrounding residues likely has other impacts on CHMP2b such as disruption of its intracellular trafficking, with potentially detrimental effects on subcellular compartments that depend on long-

[1]Neurobiology and Behavior PhD Program, Columbia University, New York, NY, USA [2]Department of Pathology and Cell Biology, Columbia University Medical Center, New York, NY, USA [3]Department of Neuroscience, Columbia University, New York, NY, USA

Correspondence: cw2622@cumc.columbia.edu

distance protein and organelle transport, such as neuronal synapses. Indeed, recent studies of a CHMP2b[intron5] transgenic mouse model have demonstrated that this mutation impairs synaptic vesicle (SV) cycling and neurotransmitter release and promotes the accumulation of proteins and endosomal structures within presynaptic boutons (Clayton et al, 2022; Waegaert et al, 2022). Whether these defects are caused by CHMP2b[intron5]-mediated disruption of SV recycling and endolysosomal trafficking at synapses, or CHMP2b[intron5] disruption of other trafficking events, for example, the axonal transport of WT CHMP2b and/or other proteins that facilitate SV protein degradation, remains unknown.

In our previous work, we showed that the ESCRT pathway mediates the activity-dependent degradation of SV membrane proteins, catalyzed by ESCRT-0 protein Hrs (Birdsall et al, 2022). In particular, we found that Hrs undergoes activity-dependent transport along axons to presynaptic boutons and that disruption of this transport impairs the degradative sorting of specific SV proteins (Birdsall et al, 2022). Whether the transport of CHMP2b, the final component of the ESCRT pathway, exhibits similar properties has not been explored. In this study, we use live imaging to characterize the axonal transport dynamics of CHMP2b and determine whether and how they are disrupted by the CHMP2b[intron5] mutation. We find that CHMP2b is recruited to synapses in an activity- and Hrs-dependent manner. Moreover, CHMP2b is transported on acidic vesicles with other ESCRT proteins and the late endosome (LE) markers Rab7 and LAMP1, consistent with its reported LE/MVB localization. In contrast, CHMP2b[intron5] exhibits little processive movement or synaptic localization at baseline, and these features are insensitive to neuronal activity. Instead, CHMP2b[intron5] vesicles exhibit oscillatory behavior reminiscent of a tug-of-war between kinesin and dynein motor proteins, as well as significantly decreased acidification. We show that these defects are due to the reduced binding of CHMP2b[intron5] to kinesin–binding protein (KBP), an inhibitor of kinesin-mediated transport, leading to dysregulation of the mutant's axonal trafficking and recruitment to synapses. Taken together, this study provides novel insights into how the trafficking and synaptic localization of CHMP2b are regulated and how the FTD-linked CHMP2b[intron5] mutation disrupts these events.

# Results

### CHMP2b recruitment to synapses is regulated by neuronal activity and Hrs

We previously showed that Hrs is recruited to presynaptic boutons in an activity-dependent manner and that this recruitment plays a key role in the degradation of SV proteins (Birdsall et al, 2022). To determine whether CHMP2b also localizes to presynaptic boutons, and if so whether this localization is regulated by neuronal activity, we measured the colocalization of mCh-CHMP2b with the synaptic vesicle (SV)–associated protein EGFP-Synapsin1a in 14 day in vitro (DIV) hippocampal neurons. We used Synapsin1a as a presynaptic marker based on its specific accumulation at boutons and lack of vesicular axonal transport (Tang et al, 2013), thereby providing us

with a clear readout of CHMP2b's presynaptic localization rather than its cotransport with SV-associated proteins. EGFP-Synapsin1a was coexpressed with a control shRNA (shCtrl) to facilitate comparison with Hrs knockdown experiments described in the next section. Before fixation and imaging, neurons were treated for 2 h with vehicle control (DMSO) or the GABA receptor antagonist bicuculline and the voltage-activated potassium channel blocker 4-aminopyridine (BIC/4AP) to elicit neuronal firing, as in our previous studies (Sheehan et al, 2016; Birdsall et al, 2022). These experiments revealed that mCh-CHMP2b has a punctate expression pattern and that ~65% of mCh-CHMP2b puncta colocalize with EGFP-Synapsin1a at baseline (Fig 1A and E), similar to the percentage of Hrs colocalization with Synapsin1a (Sheehan et al, 2016; Birdsall et al, 2022). This value was significantly increased by neuronal activity, with more than 80% of mCh-CHMP2b puncta colocalizing with EGFP-Synapsin1a after BIC/4AP treatment (Fig 1B and E). To verify that overexpressed CHMP2b reflects the behavior of the endogenous protein, we treated 14 DIV hippocampal neurons with vehicle and BIC/4AP as above, then immunostained with antibodies against endogenous CHMP2b and vesicle-associated membrane protein 2 (VAMP2) to label presynaptic boutons. We found that endogenous CHMP2b exhibits a punctate expression pattern similar to that of mCh-CHMP2b (Fig S1A and B) and that BIC/4AP treatment increased CHMP2b colocalization with VAMP2 compared with the control condition (Fig S1A–C), consistent with our findings for mCh-CHMP2b.

Because Hrs is the initiating component of the ESCRT pathway (Raiborg & Stenmark, 2009), we next investigated whether this ESCRT-0 protein is required for the synaptic recruitment of CHMP2b. We performed the same experiment as above, using a construct that coexpresses EGFP-Synapsin1a and an shRNA to knock down Hrs (shHrs; Fig S1D and E). Here, shHrs significantly decreased CHMP2b colocalization with EGFP-Synapsin1a compared with shCtrl, both at baseline (from ~65% to 40%; Fig 1C and E) and in response to BIC/4AP treatment (from ~80% to 60%; Fig 1D and E). Notably, shHrs expression did not alter the number of EGFP-Synapsin1a or mCh-CHMP2b puncta per unit length of the axon (Fig 1F and G), indicating that these findings are not due to an overall loss of presynaptic boutons or a decrease in CHMP2b axonal localization, but rather to a key role of Hrs in the synaptic recruitment of CHMP2b. Moreover, Hrs knockdown did not prevent the ~20% increase in mCh-CHMP2b colocalization with EGFP-Synapsin1a after BIC/4AP treatment, demonstrating that CHMP2b can still undergo activity-dependent recruitment to synapses under conditions of Hrs depletion (Fig 1D and E).

### Neuronal activity increases the retrograde transport of CHMP2b and its cotransport with other ESCRT proteins

Our finding that neuronal activity regulates the synaptic recruitment of CHMP2b suggests that its axonal transport is also regulated by activity. To test this, we analyzed the transport dynamics of EGFP-CHMP2b vesicles in 14 DIV neurons at baseline and after 2-h treatment with BIC/4AP. Most of the vesicles (~60%) were stationary in both conditions (Fig 2A–C). Of the motile vesicles, most exhibited nonprocessive bidirectional motility under control conditions, with smaller proportions exhibiting anterograde or retrograde movement (Fig 2A and C; Video 1). Treatment with BIC/4AP nearly doubled

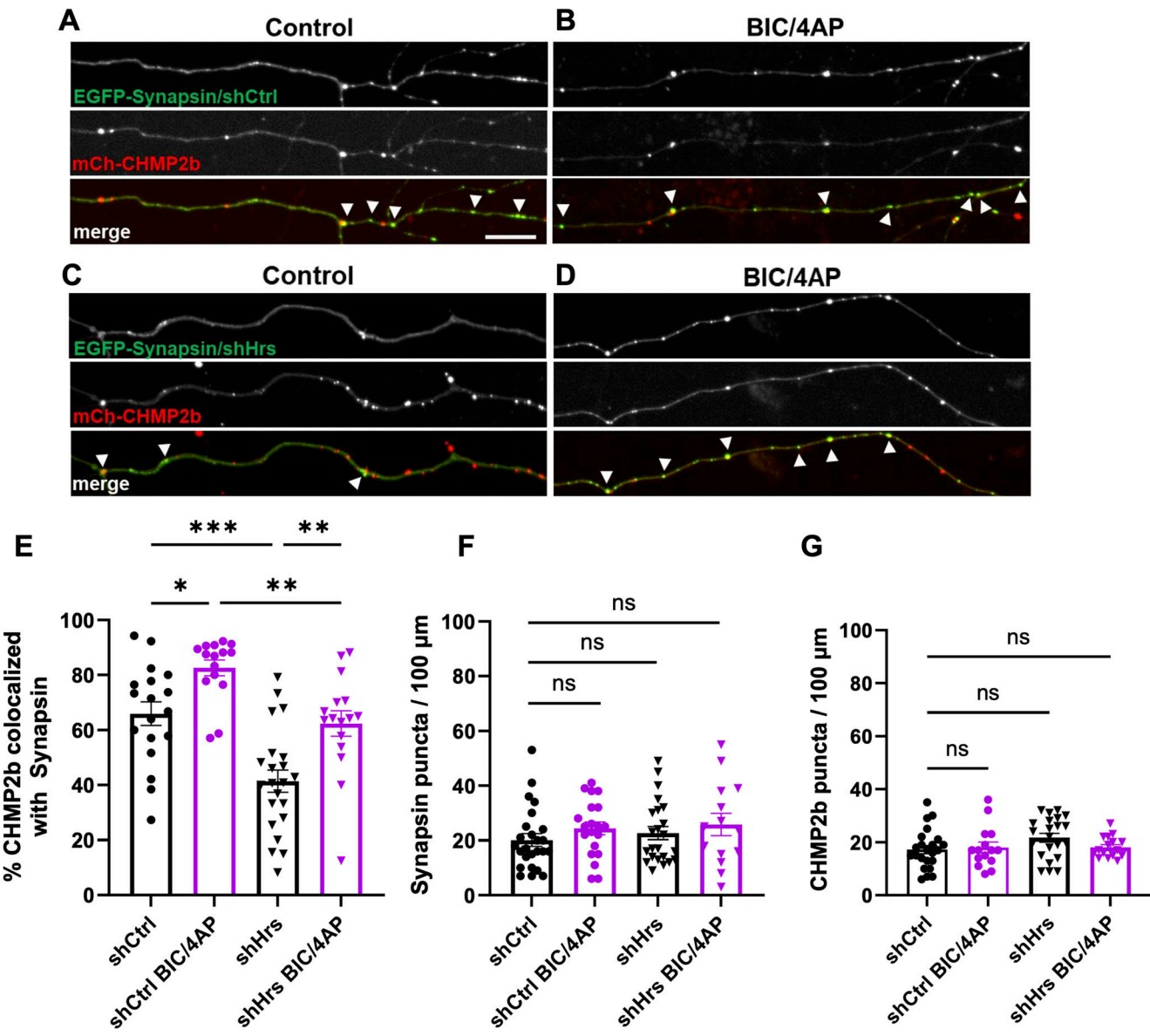

**Figure 1. CHMP2b recruitment to synapses is regulated by neuronal activity and Hrs.**
**(A, B)** Single-channel and merged images of 14 DIV hippocampal axons expressing mCh-CHMP2b and EGFP-Synapsin1a together with a control shRNA (shCtrl) and treated for 2 h with DMSO (control; (A)) or bicuculline and 4-aminopyridine (BIC/4AP; (B)). **(C, D)** Single-channel and merged images of 14 DIV hippocampal axons expressing mCh-CHMP2b and EGFP-Synapsin1a together with an shRNA to knockdown Hrs (shHrs), under control (C) or BIC/4AP (D) conditions. **(E)** Percentage of CHMP2b puncta that colocalize with Synapsin puncta (*$P$ = 0.0359, **$P$ = 0.0091, **$P$ = 0.0023, ***$P$ = 0.0002, one-way ANOVA with Tukey's multiple comparisons test; n = 18 [shCtrl], 15 [shCtrl + BIC/4AP], 22 [shHrs], 16 [shHrs + BIC/4AP] axons/condition; N ≥ 3 independent experiments). **(F)** Average number of EGFP-Synapsin puncta per 100 $\mu$m of axon (ns; one-way ANOVA with Tukey's multiple comparisons test; n = 25 [shCtrl], 20 [shCtrl + BIC/4AP], 23 [shHrs], 14 [shHrs + BIC/4AP] axons/condition; N ≥ 3 independent experiments). **(G)** Average number of mCh-CHMP2b puncta per 100 $\mu$m of axon (ns; one-way ANOVA with Tukey's multiple comparisons test; n = 23 [shCtrl], 15 [shCtrl + BIC/4AP], 22 [shHrs], 16 [shHrs + BIC/4AP] axons/condition; N ≥ 3 independent experiments). For all images, size bar = 10 $\mu$m. White arrowheads indicate CHMP2b-Synapsin colocalization. All scatter plots show the mean ± SEM.

the number of motile EGFP-CHMP2b vesicles, from ~28% to ~54% (Fig 2B and D; Video 2), similar to what we previously observed for Hrs (Birdsall et al, 2022). Directional movement was significantly increased in the retrograde category only (from 10% to 20% of vesicles), whereas bidirectional and anterograde transport both trended toward decreasing (Fig 2B and C). Conversely, blocking neuronal firing with the voltage-gated sodium channel blocker tetrodotoxin (TTX) caused a significant decrease (~33%) in CHMP2b

motility (Fig S1F–H), further demonstrating that its transport is regulated by neuronal activity. We also observed that CHMP2b vesicles had an average displacement velocity of 0.31 $\mu$m/sec and displacements between 4 and 20 $\mu$m, with no difference between control and BIC/4AP conditions (Fig 2E–G). This profile is in notable contrast to that of Hrs transport vesicles, which display increased bidirectional and anterograde motility in response to activity (Birdsall et al, 2022). These findings indicate that neuronal firing

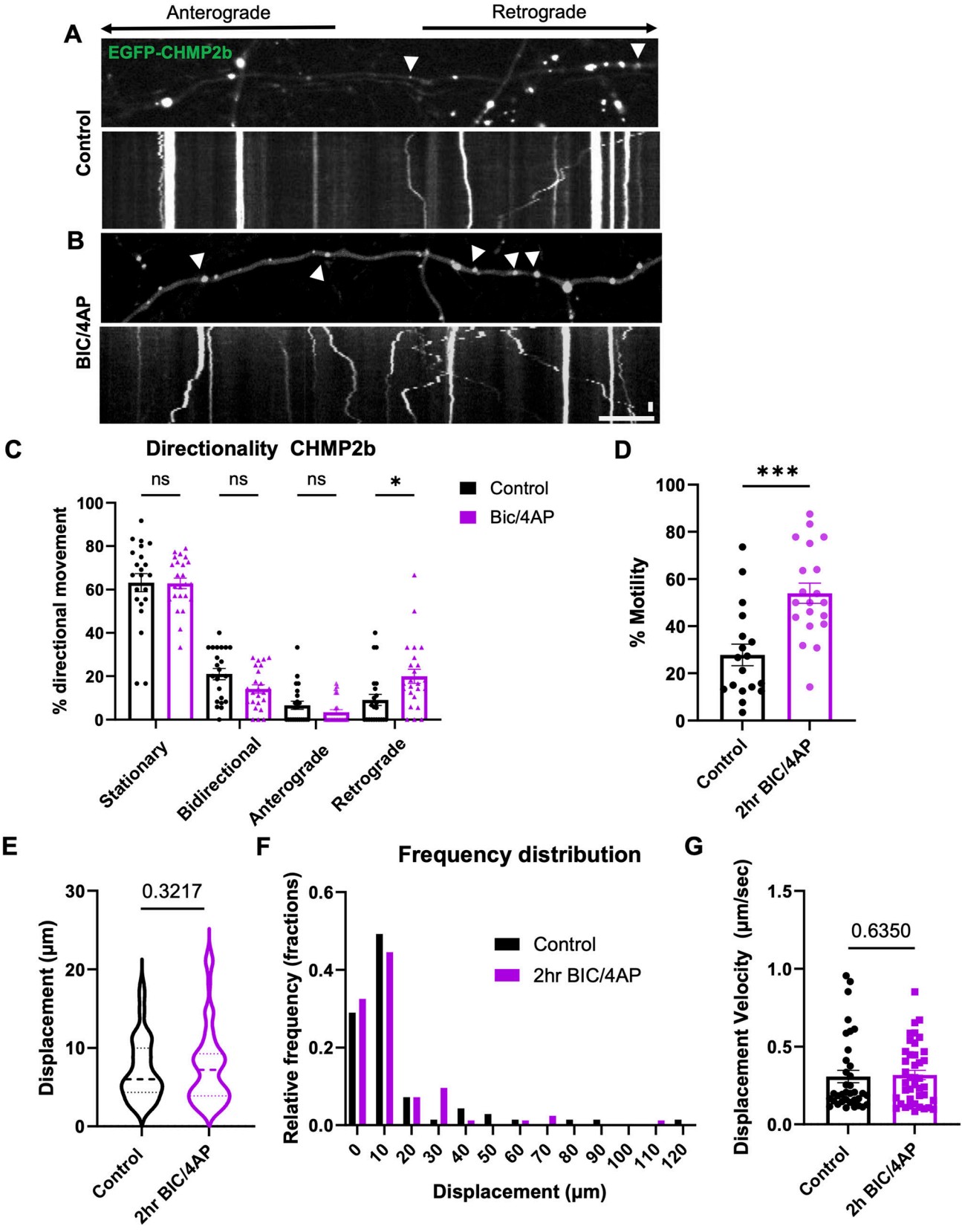

stimulates the transport of both ESCRT-0 and ESCRT-III proteins, but in opposite directions.

Although Hrs and CHMP2b exhibit differences in their axonal transport dynamics, both proteins are recruited to synapses in response to neuronal activity, suggesting that at least a fraction of these ESCRT proteins may be cotransported. To investigate this issue, we examined the colocalization and cotransport of mCh-Hrs and EGFP-CHMP2b in axons of 14 DIV neurons. We observed relatively high colocalization of CHMP2b with Hrs, and vice versa, in fixed axons after treatment with vehicle control (40% and 60%, respectively; Fig 3A–D), with 2-h BIC/4AP treatment further increasing their colocalization (to 60% and 80%, respectively; Fig 3A–D). Similarly, in live cells, ~25% of motile vesicles carried both mCh-Hrs and EGFP-CHMP2b, and this percentage increased to nearly 60% after BIC/4AP treatment (Fig 3E–G). Intriguingly, vesicles positive for both proteins exhibited a retrograde bias (Fig 3H), indicating that the presence of CHMP2b drives their directional motility.

To determine whether other ESCRT-III proteins are also cotransported with CHMP2b, we measured the colocalization and cotransport of EGFP-CHMP2b with another CHMP protein, mCh-CHMP4b, in fixed axons and during live-imaging sessions. Similar to our findings with Hrs, we observed that a high proportion of CHMP2b puncta colocalized with CHMP4b, and vice versa, in fixed axons at baseline (40% and 60%, respectively; Fig 3I–L). However, although CHMP4b colocalization with CHMP2b increased after BIC/4AP treatment, the converse was not true, as neuronal activity did not significantly alter CHMP2b colocalization with CHMP4b (Fig 3I–L). This discrepancy may be related to CHMP2b's essential role in terminating the growing chain of CHMP4b filaments during MVB formation (Teis et al, 2008), which necessitates CHMP2b recruitment to CHMP4b+ structures, but not vice versa. On the contrary, in live-imaging experiments, BIC/4AP treatment significantly increased the proportion of motile CHMP2b puncta cotransported with CHMP4b (from 30 to 40%; Fig 3M–O), with most of these vesicles moving bidirectionally or in the retrograde direction (Fig 3P). These data indicate that the cotransport of these ESCRT-III proteins is regulated by activity.

## CHMP2b[intron5] vesicles exhibit impaired axonal transport, maturation, and recruitment to synapses

Recent studies have shown that the FTD-causative CHMP2b[intron5] mutation disrupts SV recycling and promotes the accumulation of endosomal structures in presynaptic boutons of transgenic mice (Clayton et al, 2022; Waegaert et al, 2022). To investigate whether these deficits could result from impaired axonal transport of CHMP2b[intron5], we expressed CHMP2b terminated at residue 178, a methionine-to-valine mutation (M178V), which represents the major protein product of this intronic mutation (Fig 4A), in 14 DIV hippocampal neurons. Our live-imaging experiments revealed that EGFP-CHMP2b[intron5] has a similar punctate expression pattern, axonal distribution, and proportion of WT CHMP2b (Fig 4B–D). However, EGFP-CHMP2b[intron5] puncta exhibited a strikingly different type of motility compared with WT CHMP2b puncta. Specifically, although we observed rare instances of EGFP-CHMP2b[intron5] vesicles undergoing processive directional movement, the majority exhibited rapid back-and-forth movements reminiscent of a "tug-of-war" between dynein and kinesin motor proteins (Fig 4B and C; Video 3). We classified these movements as "oscillatory" rather than bidirectional because nearly all of them were less than 4 μm in a given direction, our criterion for bidirectional movement (Fig 4D; see the Materials and Methods section for details of motility parameters). In addition, the motility of EGFP-CHMP2b[intron5] vesicles was insensitive to BIC/4AP treatment (Fig 4C–E; Video 4), demonstrating that CHMP2b[intron5] axonal transport is not regulated by neuronal activity. Because *CHMP2b[intron5]* is an autosomal dominant mutation, we further investigated whether CHMP2b[intron5] interferes with the axonal transport of WT CHMP2b. Live imaging of axons coexpressing these proteins revealed that WT EGFP-CHMP2b and mCh-CHMP2b[intron5] localized primarily to the same vesicles (Fig 4F). Moreover, their transport was predominantly oscillatory, similar to that of CHMP2b[intron5] alone (Fig 4F and G; Video 5), indicating that CHMP2b[intron5] exerts a dominant-negative effect on WT CHMP2b transport.

These defects in axonal transport suggested to us that CHMP2b[intron5] vesicles may also exhibit deficits in their maturation. Because CHMP2b is typically regarded as a marker of MVBs/LEs, we assessed its cotransport with the late endosome markers Rab7 and LAMP1 (Zerial & McBride, 2001; Cheng et al, 2018), and the acidity of CHMP2b+ vesicles with LysoTracker Red. Here, we found that ~40% of motile EGFP-CHMP2b vesicles carried mCh-Rab7 or mCh-LAMP1 (Fig 5A, B, G, and H) and that 73% were labeled with LysoTracker Red (Fig 5C and I), suggesting that most of the CHMP2b vesicles represent acidified endosomal structures. Although a similar proportion of motile CHMP2b[intron5] vesicles were LAMP1+ (Fig 5E and H), significantly fewer were Rab7+ (30% versus. 40% for WT; Fig 5D and G), and only 37% were labeled with LysoTracker Red (Fig 5F and I), representing a sizable reduction in the proportion of acidified structures compared with WT CHMP2b. Collectively, these results suggest that CHMP2b[intron5] vesicles have defects in their axonal transport and maturation.

The observed deficits in CHMP2b[intron5] transport indicate that its recruitment to synapses is impaired. To test this prediction, we cotransduced neurons with mCh-CHMP2b[intron5] and either EGFP-

---

**Figure 2. Neuronal activity induces retrograde transport of CHMP2b in axons.**
**(A, B)** Still images and kymographs of 14 DIV hippocampal axons expressing EGFP-CHMP2b, representing a 4-min imaging period, under control (A) and BIC/4AP (B) conditions. The direction of movement is indicated. **(C)** Percentage of EGFP-CHMP2b puncta from stationary, bidirectional, anterograde, or retrograde categories of movement (*$P = 0.0145$, one-way ANOVA with Dunn's multiple comparisons test; n = 22 [control], 24 [BIC/4AP] axons/condition; N ≥ 3 independent experiments). **(D)** Percentage of motile EGFP-CHMP2b puncta at baseline and after 2-h BIC/4AP treatment (***$P = 0.0002$, unpaired t test; n = 18 [control], 20 [BIC/4AP] axons/condition; N ≥ 3 independent experiments). **(E, F, G)** Distributions of EGFP-CHMP2b total displacements under control and BIC/4AP conditions (E, F) (ns, $P = 0.3217$, Kolmogorov–Smirnov test; n = 69 [control], 83 [BIC/4AP] events/condition; N ≥ 3 independent experiments) and their displacement velocities (G) (ns, $P = 0.6350$, Mann–Whitney U test; n = 69 [control], 83 [BIC/4AP] events/condition; N ≥ 3 independent experiments). For all images, horizontal size bar = 10 μm, vertical scale bar = 30 s. White arrowheads indicate motile EGFP-CHMP2b puncta. All scatter plots show the mean ± SEM.

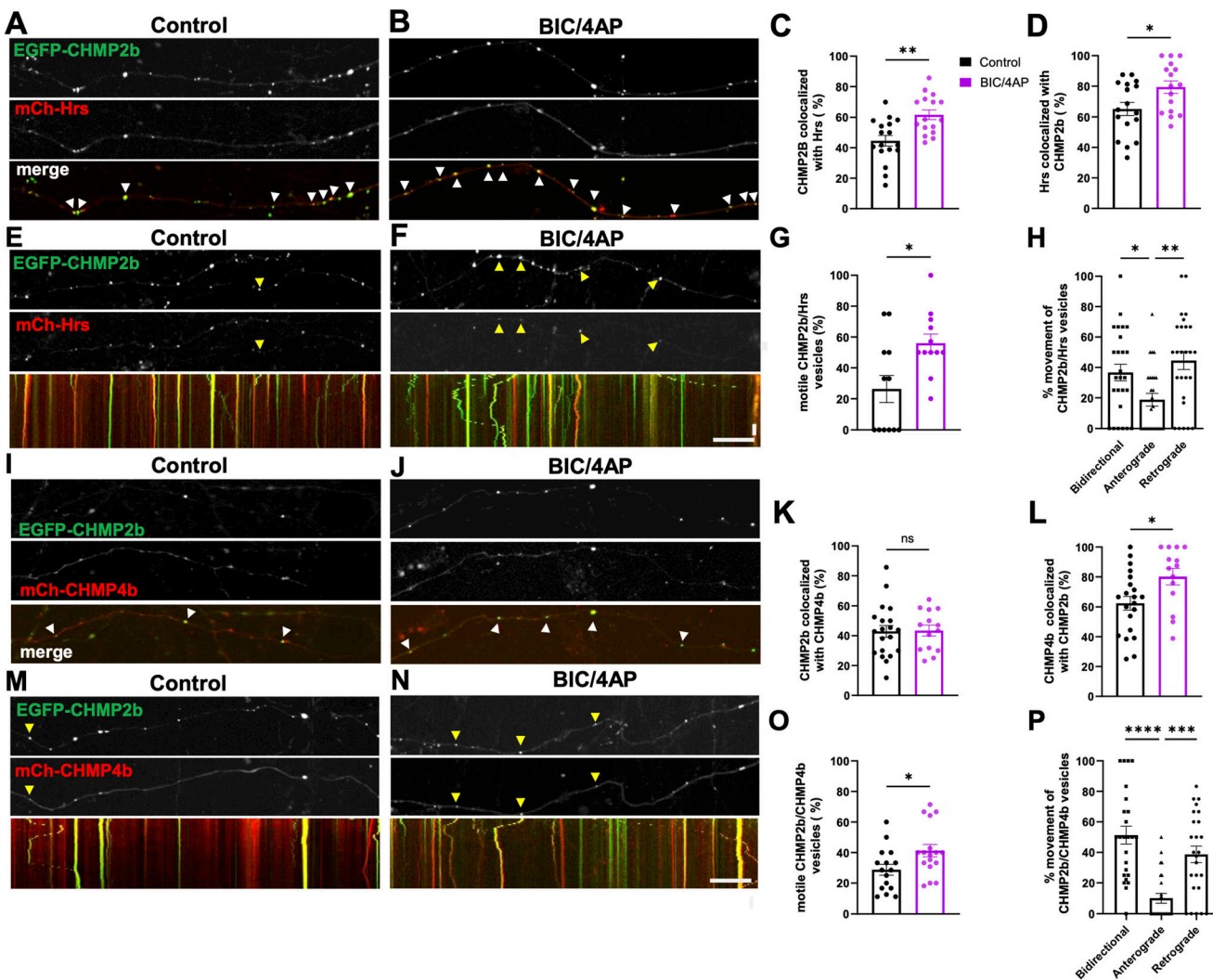

**Figure 3. CHMP2b is cotransported with other ESCRT proteins in response to neuronal activity.**
**(A, B)** Single-channel and merged images of 14 DIV hippocampal axons expressing EGFP-CHMP2b and mCh-Hrs after control (DMSO) (A) or 2-h BIC/4AP treatment (B). **(C)** Percentage of EGFP-CHMP2b puncta that colocalize with mCh-Hrs puncta (**$P = 0.0011$, unpaired $t$ test; n = 17 [shCtrl], 16 [BIC/4AP] axons/condition; N ≥ 3 independent experiments). **(D)** Percentage of mCh-Hrs puncta that colocalize with EGFP-CHMP2b puncta (*$P = 0.0226$, $t$ test; n = 17 [shCtrl], 16 [BIC/4AP] axons/condition; N ≥ 3 independent experiments). **(E, F)** Still images and kymographs of 14 DIV hippocampal axons coexpressing EGFP-CHMP2b and mCh-Hrs, representing a 4-min imaging period, under control (E) or BIC/4AP conditions (F). **(G)** Percentage of motile vesicles copositive for EGFP-CHMP2b and mCh-Hrs (*$P = 0.0102$, Mann–Whitney $U$ test; n = 12 [control], 12 [2-h BIC/4AP] axons/condition; N = 3 independent experiments). **(H)** Percentage of motile vesicles copositive for CHMP2b and Hrs that exhibit bidirectional, anterograde, or retrograde movement after 2-h treatment with BIC/4AP (**$P = 0.0025$, *$P = 0.0477$, one-way ANOVA with Tukey's multiple comparisons test; n = 26 axons/condition; N ≥ 3 independent experiments). **(I, J)** Single-channel and merged images of 14 DIV hippocampal axons expressing EGFP-CHMP2b and mCh-CHMP4b after control (I) or 2-h BIC/4AP treatment (J). **(K)** Percentage of EGFP-CHMP2b puncta that colocalize with mCh-CHMP4b puncta (ns, $P = 0.9260$, unpaired $t$ test; n = 21 [shCtrl], 14 [BIC/4AP] axons/condition; N ≥ 3 independent experiments). **(L)** Percentage of mCh-CHMP4b puncta that colocalize with EGFP-CHMP2b puncta (*$P = 0.0186$, unpaired $t$ test; n = 21 [shCtrl], 14 [BIC/4AP] axons/condition; N ≥ 3 independent experiments). **(M, N)** Still images and kymographs of 14 DIV hippocampal axons coexpressing EGFP-CHMP2b and mCh-CHMP4b, representing a 4-min imaging period, under control (M) or BIC/4AP conditions (N). **(O)** Percentage of motile vesicles copositive for EGFP-CHMP2b and mCh-CHMP4b (*$P = 0.0277$, unpaired $t$ test; n = 16 [control], 17 [BIC/4AP] axons/condition; N = 3 independent experiments). **(P)** Percentage of motile vesicles copositive for EGFP-CHMP2b and mCh-CHMP4b that exhibit bidirectional, anterograde, or retrograde movement after 2-h treatment with BIC/4AP (****$P < 0.0001$, ***$P = 0.0003$, one-way ANOVA with Tukey's multiple comparisons test; n = 25 axons/condition; N ≥ 3 independent experiments). Yellow arrowheads indicate motile vesicles cotransporting EGFP-CHMP2b and mCh-Hrs or mCh-CHMP4b. For all images, horizontal size bar = 10 μm, vertical scale bar = 30 s. White arrowheads indicate colocalization of EGFP-CHMP2b with mCh-Hrs or mCh-CHMP4b in fixed neurons. All scatter plots show the mean ± SEM.

Synapsin1a/shCtrl or EGFP-Synapsin1a/shHrs as described previously (Fig 1), and treated 14 DIV neurons with vehicle or BIC/4AP. Under baseline conditions (shCtrl and vehicle), only ~27% of mCh-CHMP2b[intron5] puncta colocalized with EGFP-Synapsin1a (Fig 6A and E), a dramatic reduction compared with the ~65% of WT mCh-

CHMP2b puncta (Figs 1 and 6E). Moreover, treatment with BIC/4AP did not increase the colocalization of mCh-CHMP2b[intron5] with EGFP-Synapsin1a, in contrast to WT mCh-CHMP2b (Figs 1 and 6B and E), indicating that this mutant does not undergo activity-dependent recruitment to synapses. Similarly, knockdown of Hrs

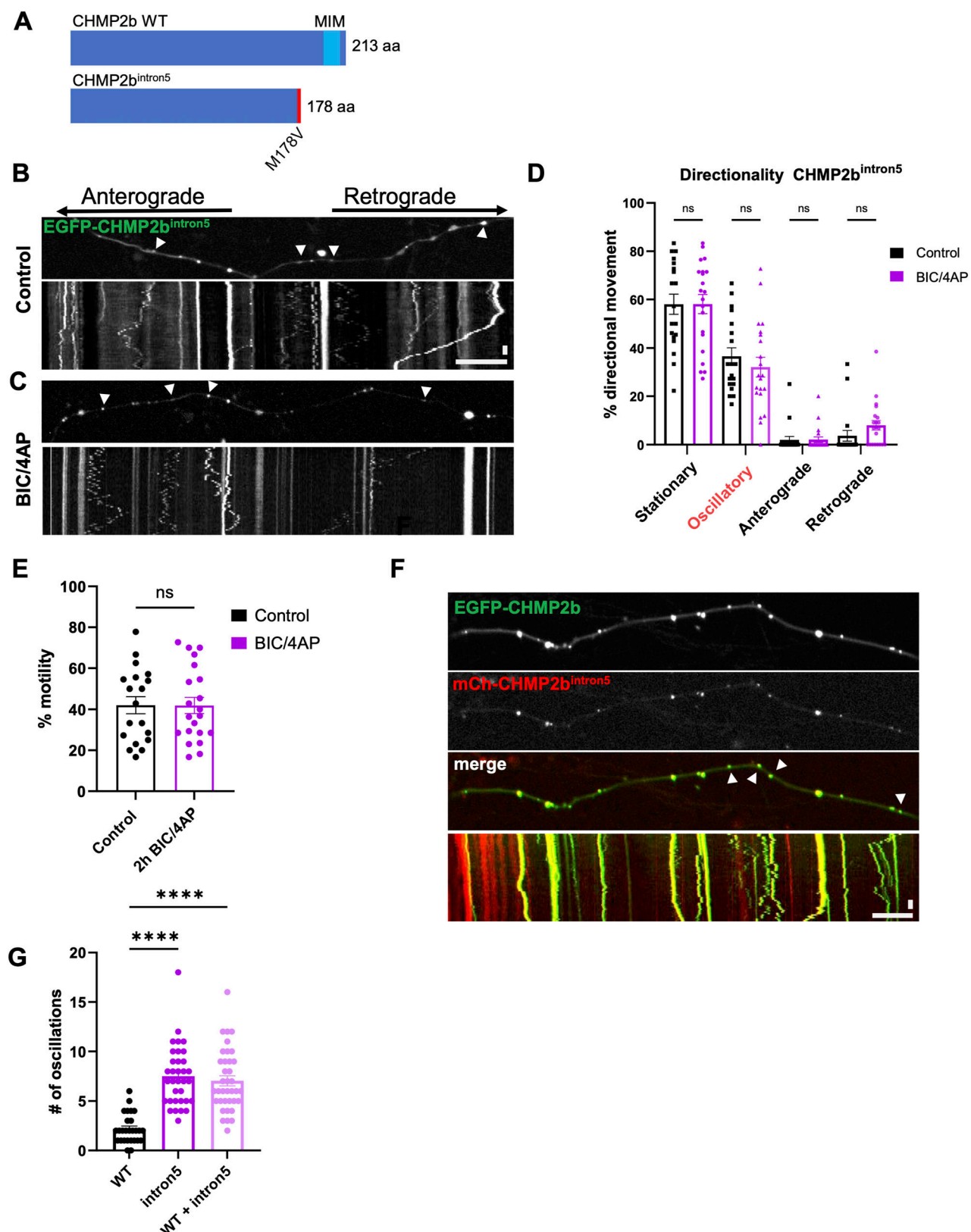

**Figure 4. CHMP2b**intron5 **exhibits aberrant oscillatory movement.**
**(A)** Schematic of WT CHMP2b and CHMP2b intron5 gene products, highlighting the location of the MIM domain and the M178V mutation. **(B, C)** Still images and associated kymographs of 14 DIV hippocampal axons expressing EGFP-CHMP2b intron5, representing a 4-min imaging period, under both control (B) and BIC/4AP conditions (C).

did not impact mCh-CHMP2b^intron5 colocalization with EGFP-Synapsin1a under control or BIC/4AP conditions (Fig 6C–E), or mCh-CHMP2b^intron5 axonal distribution (Fig 6F), demonstrating that CHMP2b^intron5 localization and recruitment to synapses are insensitive to both neuronal activity and Hrs levels. Finally, we found that CHMP2b^intron5 also acts as a dominant-negative mutant in determining the synaptic localization of WT CHMP2b. Specifically, the coexpression of mCh-CHMP2b^intron5 led to a 50% decrease in WT EGFP-CHMP2b localization to presynaptic boutons (based on colocalization with endogenous VAMP2) compared with the expression of EGFP-CHMP2b alone, and prevented the activity-dependent increase in WT CHMP2b presynaptic recruitment (Fig 6G–K). Together, these data show that CHMP2b^intron5 induces aberrant axonal transport of WT CHMP2b that interferes with its localization and recruitment to synapses.

### Kinesin-binding protein (KBP) interacts with WT CHMP2b but not CHMP2b^intron5

The aberrant transport of CHMP2b^intron5 indicates that its interactions with kinesin and/or dynein motor proteins are altered. Using the BioGRID database (https://thebiogrid.org/interaction/3222435), we queried the interactome of CHMP2b to find potential kinesin- and dynein-related proteins. We identified KIAA1279, also called kinesin-binding protein (KBP), as a putative interacting partner of CHMP2b but not any other ESCRT proteins. KBP has been shown to inhibit the microtubule association of specific kinesin family members by interacting with their motor domains (Kevenaar et al, 2016; Solon et al, 2021). We postulated that disrupted interactions between KBP and CHMP2b^intron5 could alter the balance of kinesin- versus dynein-directed transport of this CHMP2b mutant, leading to the observed "tug-of-war" behavior. To test this hypothesis, we first evaluated the binding interactions between KBP and WT or mutant CHMP2b by performing coimmunoprecipitation (IP) experiments in N2a cells cotransfected with EGFP-KBP and either mCherry, mCh-CHMP2b, or mCh-CHMP2b^intron5. We found that IP of EGFP-KBP pulled down significantly more coexpressed mCh-CHMP2b versus coexpressed mCh-CHMP2b^intron5 or mCherry alone (Fig 7A and B). These findings support an interaction between KBP and CHMP2b that is significantly reduced by the CHMP2b^intron5 mutation. We further investigated this KBP-CHMP2b interaction in situ using the proximity ligation assay (PLA) in hippocampal neurons transfected with mCh-CHMP2b or mCh-CHMP2b^intron5. After fixation and immunolabeling, PLA was performed to visualize interactions between mCh-tagged WT or mutant CHMP2b and endogenous KBP in neuronal cell bodies. In validation of our IP results, we observed significantly higher PLA punctum density in neurons expressing WT CHMP2b compared to CHMP2b^intron5, with no

background signal from the expression of mCherry alone (Fig 7C–E). Because CHMP2b transport is regulated by neuronal activity, we further tested whether the CHMP2b-KBP interaction was increased by 2-h treatment with BIC/4AP. Although BIC/4AP led to a highly significant increase in the density of PLA puncta for WT mCh-CHMP2b– versus mCh-CHMP2b^intron5–expressing neurons (Fig 7C–F), this treatment did not significantly increase the number of puncta in WT mCh-CHMP2b–expressing neurons compared with the control condition (although the data trend in this direction; Fig 7F). Together, these findings indicate that KBP differentially interacts with WT CHMP2b and CHMP2b^intron5 and that reduced interaction between KBP and CHMP2b^intron5 could underlie the axonal transport deficits of CHMP2b^intron5.

### KBP gain-of-function and loss-of-function impact the axonal transport of WT CHMP2b but not CHMP2b^intron5

We next tested whether KBP overexpression alters WT CHMP2b axonal transport, as we predicted based on their binding interaction. Here, WT mCh-CHMP2b was coexpressed with EGFP (control) or EGFP-KBP in hippocampal neurons, which were subsequently treated with vehicle or BIC/4AP as described previously. In the presence of EGFP, we found that WT mCh-CHMP2b vesicles displayed similar transport profiles as previously observed (Fig 2), with a significant decrease in bidirectional motility and an increase in retrograde transport after BIC/4AP treatment (Fig 8A–C). The coexpression of EGFP-KBP abolished the processive movement of mCh-CHMP2b vesicles observed at baseline and induced similar oscillatory behavior to that seen for CHMP2b^intron5 (Fig 8D and F), indicative of a regulatory role of KBP in CHMP2b transport. This effect of KBP was not due to its global disruption of axonal transport, as the coexpression of EGFP-KBP with mCh-Rab5 (localized to a different population of vesicles from CHMP2b) did not alter Rab5's transport profile, which included a mixture of anterograde and bidirectional movement (Fig S2A–C) as previously reported (Goto-Silva et al, 2019; Birdsall et al, 2022). Notably, BIC/4AP treatment still induced the retrograde transport of mCh-CHMP2b in the presence of EGFP-KBP (Fig 8E and F), demonstrating that KBP overexpression does not inhibit the activity-dependent increase in CHMP2b motility. Because our binding data show reduced interaction between KBP and CHMP2b^intron5, indicating that KBP overexpression should have minimal impact on the transport of CHMP2b^intron5, we also tested this concept in neurons coexpressing EGFP or EGFP-KBP together with mCh-CHMP2b^intron5. Indeed, CHMP2b^intron5 transport dynamics were unaffected by EGFP-KBP under both control and BIC/4AP conditions (Fig 8G–L).

Finally, we assessed the transport of CHMP2b in axons after KBP knockdown, which was also hypothesized to disrupt WT-CHMP2b

---

**(D)** Percentage of EGFP-CHMP2b^intron5 puncta from stationary, bidirectional, anterograde, or retrograde categories of movement under control or BIC/4AP conditions (ns, Kruskal–Wallis test with Dunn's multiple comparisons; n = 19 [control], 21 [BIC/4AP] axons/condition; N ≥ 3 independent experiments). **(E)** Percentage of motile EGFP-CHMP2b^intron5 puncta at baseline and after 2-h BIC/4AP treatment (ns, unpaired t test; n = 19 [control], 21 [BIC/4AP] axons/condition; N ≥ 3 independent experiments). **(F)** Still images and associated kymographs of 14 DIV hippocampal axons expressing mCh-CHMP2b^intron5 together with WT EGFP-CHMP2b. **(G)** Number of oscillations in axons that overexpress WT EGFP-CHMP2b, mutant mCh-CHMP2b^intron5, or both WT and mutant CHMP2b (****$P$ < 0.00010.0057, one-way ANOVA with Dunn's multiple comparisons test; n = 34 [WT], 27 [intron5], and 37 [WT + intron5] axons/condition; N ≥ 3 separate experiments). For all images, horizontal size bar = 10 $\mu$m, vertical scale bar = 30 s. White arrowheads indicate motile EGFP-CHMP2b^intron5 or EGFP-CHMP2b puncta. All scatter plots show the mean ± SEM.

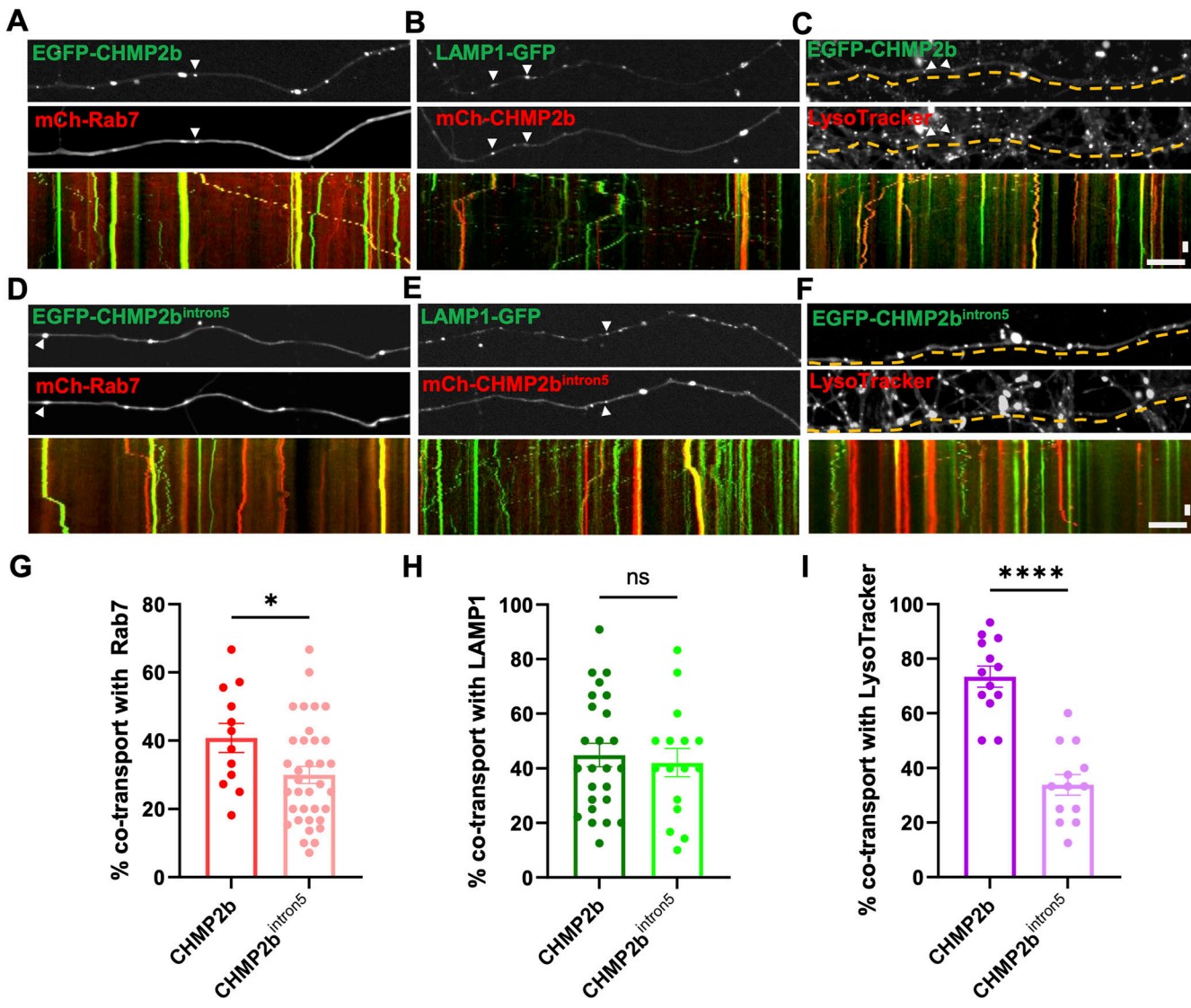

**Figure 5.   CHMP2b^intron5 vesicles display defects in maturation.**
**(A, B, C)** Still images and kymographs of 14 DIV hippocampal axons coexpressing EGFP/mCh-CHMP2b with either mCh-Rab7 (A) or LAMP1-GFP (B), or in the presence of LysoTracker Red (C), representing a 4-min imaging period. **(D, E, F)** Still images and kymographs of 14 DIV hippocampal axons coexpressing EGFP/mCh-CHMP2b^intron5 with either mCh-Rab7 (D) or LAMP1-GFP (E), or in the presence of LysoTracker Red (F) representing a 4-min imaging period. **(G)** Percentage of moving EGFP-CHMP2b or EGFP-CHMP2b^intron5 vesicles that cotransport mCh-Rab7 (*P = 0.0362, unpaired t test; n = 12 [CHMP2b], 35 [CHMP2b^intron5], axons/condition; N ≥ 3 independent experiments).
**(H)** Percentage of moving mCh-CHMP2b or mCh-CHMP2b^intron5 vesicles that cotransport LAMP1-GFP (ns, unpaired t test; n = 25 [CHMP2b], 16 [CHMP2b^intron5], axons/condition; N ≥ 3 independent experiments). **(I)** Percentage of moving EGFP-CHMP2b or EGFP-CHMP2b^intron5 vesicles that cotransport LysoTracker Red (****P < 0.0001, unpaired t test; n = 13 [CHMP2b], 35 [CHMP2b^intron5], axons/condition; N ≥ 3 independent experiments) over the respective total number of colocalized vesicles. For all images, horizontal size bar = 10 μm, vertical scale bar = 30 s. White arrowheads indicate the cotransport of CHMP2b with Rab7 or LAMP1. All scatter plots show the mean ± SEM.

but not CHMP2b^intron5. Hippocampal neurons were transduced on 7 DIV with either a control shRNA or a previously characterized shRNA to knock down KBP (shKBP; [Brouwers et al, 2017]), which we found to reduce KBP protein by ~70% (Fig S2D and E). Neurons were then transfected with mCh-CHMP2b or mCh-CHMP2b^intron5 on 13 DIV, and treated with vehicle or BIC/4AP on 14 DIV before imaging. As in our previous set of experiments, we found that WT mCh-CHMP2b motility in shCtrl-expressing neurons exhibited relatively equal distributions of directional movement at baseline and an increase in retrograde movement after BIC/4AP treatment

(Fig 9A–C). Intriguingly, knockdown of KBP caused WT mCh-CHMP2b motility to mimic that of mCh-CHMP2b^intron5, with decreased processive movement, oscillatory behavior instead of bidirectional motility, and the loss of activity-induced retrograde transport (Fig 9D–F). Moreover, KBP knockdown had no impact on mCh-CHMP2b^intron5 motility in the presence or absence of neuronal activity (Fig 9G–I), as predicted given the reduced interaction between these proteins. Because KBP knockdown impaired the axonal transport of WT CHMP2b, we tested whether this knockdown would also impair CHMP2b synaptic localization. Using a

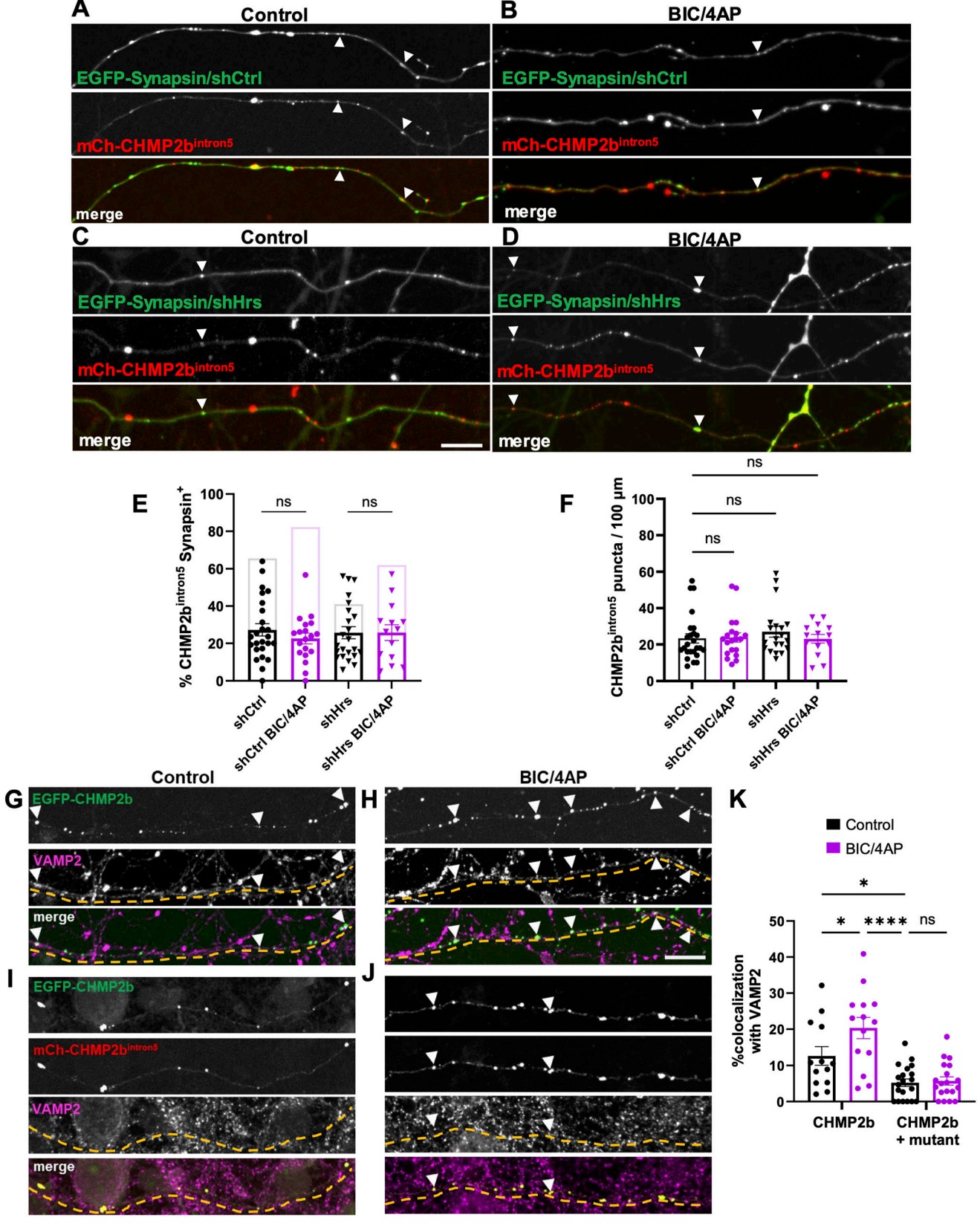

similar experimental setup as described for Fig 1, we cotransfected mCh-CHMP2b into neurons with a construct expressing EGFP-Synapsin1a and either shCtrl or shKBP, then treated the cells with either vehicle or BIC/4AP. We found that shKBP not only reduced the colocalization of mCh-CHMP2b with EGFP-Synapsin at baseline compared with shCtrl (Fig 9J and K–N), but also prevented the activity-dependent increase in mCh-CHMP2b recruitment to presynaptic boutons (Fig 9L–N). These effects were not due to a loss of presynaptic boutons or a decrease in CHMP2b axonal localization, as shKBP expression did not alter the number of EGFP-Synapsin1a or mCh-CHMP2b puncta per unit length of axon (Fig 9O and P). Together, these data provide evidence that KBP interacts with the C-terminal domain of CHMP2b to regulate its transport and synaptic localization and that this regulation is abolished in the CHMP2b[intron5] mutant, leading to the oscillatory behavior and lack of synaptic association observed for CHMP2b[intron5].

# Discussion

In the current study, we characterize the axonal transport of the ESCRT-III protein CHMP2b, the final component of the core ESCRT machinery required for ILV scission and MVB formation. We find that the motility of CHMP2b vesicles is regulated by neuronal activity, which induces their retrograde transport, promotes CHMP2b cotransport with other ESCRT proteins, and increases CHMP2b recruitment to synapses. In contrast, vesicles carrying the FTD-causative CHMP2b[intron5] mutant exhibit limited processive movement in axons, and instead display oscillatory behavior with defects in their maturation and synaptic localization/recruitment. These phenotypes are caused by deficient binding of CHMP2b[intron5] to the kinesin inhibitor KBP, demonstrating the importance of this kinesin regulatory protein for CHMP2b axonal transport.

Our findings demonstrate that the transport of CHMP2b, like Hrs, is regulated by neuronal activity. However, in contrast to the ESCRT-0 proteins Hrs and STAM1, which are transported primarily in the anterograde direction (Birdsall et al, 2022), CHMP2b undergoes primarily retrograde transport. These results are consistent with the reported directional movement of early versus late endosomes in axons (Deinhardt et al, 2006; Goto-Silva et al, 2019; Rizalar et al, 2023), and with a previous study in nonneuronal cells that observed

disparate transport dynamics for early versus late ESCRT complexes (Wenzel et al, 2018). Despite their differences in directional movement, Hrs and CHMP2b are often cotransported on the same vesicles, which exhibit the retrograde bias of CHMP2b (Fig 3). In addition, we find that Hrs is important for the synaptic recruitment of CHMP2b, as Hrs knockdown decreases CHMP2b colocalization with the presynaptic marker EGFP-Synapsin. It is likely that Hrs' role in CHMP2b synaptic recruitment is indirect and based on its ability to catalyze recruitment of the entire ESCRT pathway, including the ESCRT-I, ESCRT-II, and ESCRT-III proteins that recruit substrates, induce LE/MVB membrane curvature, and precede CHMP2b recruitment. Notably, Hrs knockdown does not prevent the activity-induced increase in CHMP2b recruitment to presynaptic boutons, indicating either that Hrs is not required for this recruitment, or that its incomplete knockdown (~70%) does not abolish this effect. Moreover, Hrs knockdown does not impact the synaptic recruitment of CHMP2b[intron5], potentially reflecting the mutant's disrupted axonal transport and therefore decreased likelihood of delivery to presynaptic boutons where it would encounter Hrs and other ESCRT proteins. However, it is also possible that CHMP2b localization to synapses requires direct binding to Hrs that is abolished in the CHMP2b[intron5] mutant. The observed cotransport of Hrs and CHMP2b, and the regulation of this event by neuronal activity, is consistent with this interesting possibility, which will be tested in future studies.

This study represents the first characterization of ESCRT-III transport dynamics in neurons. As anticipated, we find that a substantial fraction of CHMP2b is cotransported with other LE/MVB proteins, including Rab7, LAMP1, and CHMP4b. Although our reported velocity of CHMP2b vesicles (0.31 μm/s) is slower than that observed by some investigators for dynein-directed transport (Maday et al, 2014; Fu et al, 2022; Cason & Holzbaur, 2023), it fits within recently reported velocity ranges of transport (Boecker et al, 2020; Fellows et al, 2024) and exhibits characteristics, for example, rapid movements with intermittent pauses, typical of this mode of axonal transport (Roy, 2020). Moreover, CHMP2b transport speed closely parallels previously reported velocities of Rab7 and LAMP1 vesicles (Lee et al, 2011; Zhang et al, 2013; Lie et al, 2022), suggesting that many of the motile CHMP2b vesicles are LEs/MVBs undergoing retrograde transport from synapses in response to neuronal activity, reminiscent of the activity-dependent retrograde transport of amphisomes (Cheng et al, 2015; Lie et al, 2022). Additional work will

**Figure 6. Impaired synaptic localization and recruitment of CHMP2b[intron5].**
**(A, B)** Single-channel and merged images of 14 DIV hippocampal axons expressing mCh-CHMP2b[intron5] and EGFP-Synapsin1a together with a control shRNA, under control (A) or BIC/4AP conditions (B). **(C, D)** Single-channel and merged images of 14 DIV hippocampal axons expressing mCh-CHMP2b[intron5] and EGFP-Synapsin1a together with an shRNA to knockdown Hrs (shHrs), under control (C) or BIC/4AP (D) conditions. **(E)** Percentage of mCh-CHMP2b[intron5] puncta that colocalize with EGFP-Synapsin puncta (ns, one-way ANOVA with Tukey's multiple comparisons test; n = 18 [shCtrl], 15 [shCtrl + BIC/4AP], 22 [shHrs], 16 [shHrs + BIC/4AP] axons/condition; N ≥ 3 independent experiments). For comparison, the lighter bar outlines represent the average colocalization values of WT mCh-CHMP2b with EGFP-Synapsin (Fig 1E). **(F)** Average number of EGFP-Synapsin puncta per 100 μm of axon (ns, one-way ANOVA with Tukey's multiple comparisons test; n = 25 [shCtrl], 20 [shCtrl + BIC/4AP], 20 [shHrs], 14 [shHrs + BIC/4AP] axons/condition; N ≥ 3 independent experiments). **(G, H)** Single-channel and merged images of 14 DIV hippocampal axons expressing EGFP-CHMP2b and immunostained with VAMP2 antibodies, under control (G) or BIC/4AP conditions (H). **(I, J)** Single-channel and merged images of 14 DIV hippocampal axons coexpressing EGFP-CHMP2b and mCh-CHMP2b[intron5] and immunostained with VAMP2 antibodies, under control (I) or BIC/4AP conditions (J). **(K)** Percentage of CHMP2b or CHMP2b/CHMP2b[intron5] puncta colocalized with VAMP2 (*P = 0.0454, *P = 0341,****P < 0.0001, one-way ANOVA with Dunn's multiple comparisons test; n = 13 [control CHMP2b], 14 [BIC/4AP CHMP2b], 20 [control CHMP2b + mutant], 19 [BIC/4AP CHMP2b + mutant] axons/condition; N ≥ 3 independent experiments). For all images, horizontal size bar = 10 μm. White arrowheads indicate sites of CHMP2b[intron5]-Synapsin or CHMP2b/VAMP2 colocalization. Yellow dashed lines outline the axons. All scatter plots show the mean ± SEM.

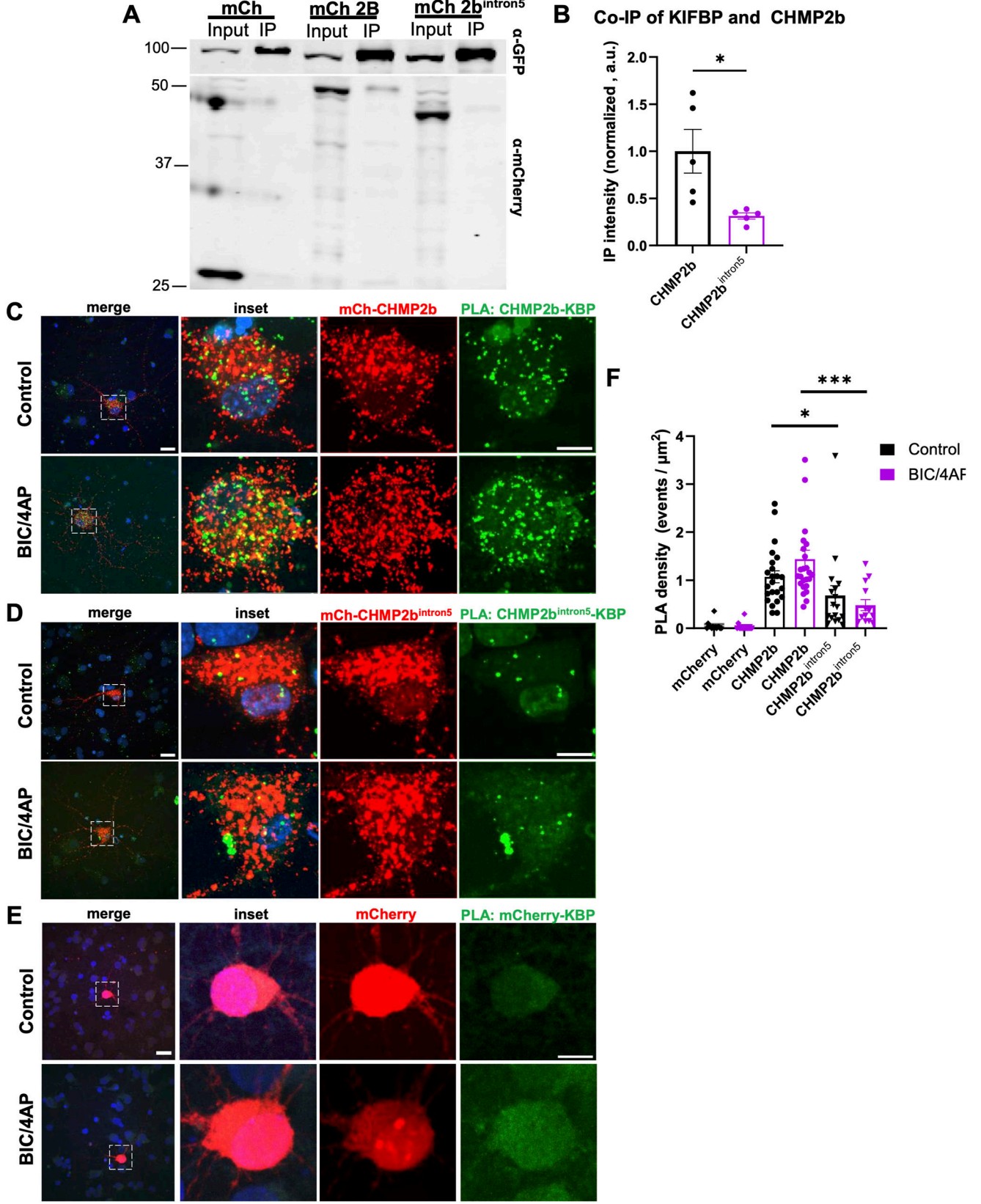

be required to characterize the morphologies and cargoes of these CHMP2b vesicles.

Intriguingly, we find that CHMP2b transport is disrupted by the CHMP2b[intron5] mutation, a causative mutation of ALS-FTD. Impaired axonal transport is implicated in a wide range of neurodegenerative disorders (Berth & Lloyd, 2023) and often precedes neuronal death (Salvadores et al, 2017). Indeed, axonal degeneration was previously reported in CHMP2b[intron5] mouse models, along with defects in presynaptic function and synaptic vesicle cycling (Ghazi-Noori et al, 2012; Clayton et al, 2022; Waegaert et al, 2022), but the mechanisms underlying these pathogenic events have remained elusive. Our findings suggest that synaptic dysfunction in CHMP2b[intron5]-expressing neurons results in part from the impaired axonal transport of WT CHMP2b, preventing the formation of MVBs (and likely also autophagosomes), and thus the proper maintenance of protein homeostasis in axons and presynaptic boutons. CHMP2b[intron5] exerts a dominant effect over WT CHMP2b, converting transport vesicles carrying both proteins to the aberrant oscillatory phenotype and preventing their synaptic localization (Figs 4 and 6). This effect parallels what is observed in human tissue of FTD-3 patients, in which CHMP2b[intron5] is present at only 35% of the levels of WT CHMP2b, yet is still sufficient to cause the disease (Urwin et al, 2010). It is not clear why WT CHMP2b transport becomes defective in the presence of the mutant, but it could be due to dimerization/aggregation of the two proteins, or to an unknown toxic gain-of-function effect caused by the CHMP2b[intron5] protein. CHMP2b[intron5] vesicles also exhibit a 25% reduction in Rab7 cotransport and a 50% reduction in LysoTracker labeling compared with WT CHMP2b vesicles (Fig 5). These data suggest that the maturation of CHMP2b[intron5] vesicles is hindered, consistent with previous reports of endolysosomal dysfunction in CHMP2b[intron5] neurons (Urwin et al, 2010; Clayton et al, 2015, 2018; Vandal et al, 2018). The cumulative consequences of these defects likely include protein accumulation/aggregation at synapses and disruption of vesicle trafficking along axons, leading to synaptic dysfunction and axon transport deficits that ultimately trigger neurodegeneration.

The oscillatory movements of CHMP2b[intron5] are indicative of its impaired interactions with motor proteins and/or their adaptors. Interestingly, CHMP2b is the only ESCRT protein in the BioGRID database reported to interact with a motor protein adaptor, in this case, the kinesin inhibitor KBP, indicative of a unique role of CHMP2b in regulating the directional transport of CHMP2b+ vesicles, many of which appear to be LEs/MVBs. Here, we validate the CHMP2b-KBP interaction and show that it is dramatically reduced or absent for CHMP2b[intron5], indicating that binding occurs through the C-terminal region of CHMP2b. We further demonstrate the relevance of this interaction for CHMP2b transport, showing that both overexpression and knockdown of KBP impede processive movement of WT CHMP2b along axons. In contrast, these manipulations have no effect on CHMP2b[intron5], as expected if the interaction with KBP is lost. Interestingly, neuronal activity stimulates the retrograde transport of WT CHMP2b under conditions of KBP overexpression but not knockdown, indicating that KBP binding to CHMP2b is required for its activity-dependent retrograde movement. Although our PLA experiments show a trend toward an activity-induced interaction between KBP and CHMP2b (Fig 7E), we were unable to see a significant effect of neuronal activity on this interaction. This finding likely reflects the fact that CHMP2b was overexpressed in these studies (to enable the comparison of mCh-CHMP2b versus mCh-CHMP2b[intron5] and bypass the lack of specificity of commercially available CHMP2b antibodies), driving high baseline levels of KBP-CHMP2b interaction that were difficult to further increase. Future studies using strategies such as CRISPR/Cas9 to introduce tagged CHMP2b constructs into the endogenous gene locus would circumvent these technical issues.

Based on our data, we propose a model wherein KBP binding to CHMP2b, likely regulated by stimuli such as neuronal activity, inhibits kinesin-mediated transport and induces the dynein-directed retrograde transport of CHMP2b vesicles. In contrast, the inability of CHMP2b[intron5] to interact with KBP prevents this negative regulation of kinesin-mediated transport, leading to the oscillatory, tug-of-war dynamics observed for CHMP2b[intron5] vesicles and inhibiting their axonal transport. Given that KBP interacts with 8 of the 45 kinesin motors, including KIF3A, KIF1A-C, KIF13B, and KIF14 (Kevenaar et al, 2016; Malaby et al, 2019), future studies are needed to determine which of these are relevant for CHMP2b transport and whether they also mediate the transport of amphisomes and other ESCRT proteins.

Altogether, these findings represent the first characterization of CHMP2b axonal transport dynamics, and implicate KBP as a key regulator of the transport of CHMP2b+ vesicles, a large fraction of which are likely LEs/MVBs. Moreover, this work provides a novel mechanistic insight into how CHMP2b[intron5] impairs the axonal transport and presynaptic recruitment of CHMP2b, contributing to our understanding of how this ALS/FTD-causative mutant induces synaptic dysfunction.

# Materials and Methods

### Primary hippocampal culture

Hippocampal neurons were obtained from E18 Sprague Dawley rat embryos of both sexes (Charles River) or purchased from a commercial source of tissue (Transnetyx) and dissociated. For live-

**Figure 7.  Kinesin-binding protein (KBP) interacts with WT CHMP2b but not CHMP2b[intron5].**
**(A)** Immunoblot of lysates from N2a cells coexpressing EGFP-KBP together with mCherry, mCh-CHMP2b, or mCh-CHMP2b[intron5], immunoprecipitated (IP) with GFP antibody, and probed with mCherry and GFP antibodies. **(B)** Coimmunoprecipitated mCherry proteins normalized to GFP control (*P = 0.0193, unpaired t test; N = 5 independent experiments). **(C, D, E)** Images of proximity ligation assay (PLA) in the soma of 14 DIV hippocampal axons expressing mCh-CHMP2b (C), mCh-CHMP2b[intron5] (D), or mCherry (E) after 2-h treatment with vehicle or BIC/4AP. PLA puncta (green) represent interactions between mCherry/mCh-tagged proteins (red) and endogenous KBP. Nuclei are indicated by DAPI staining (blue). White boxes indicate areas enlarged for inset. **(F)** PLA punctum density in soma of mCherry-, mCh-CHMP2b-, or mCh-CHMP2b[intron5]–expressing neurons (*P = 0.0476, ***P = 0.0003, Kruskal–Wallis test with multiple comparisons; n = 23 [CHMP2b], 25 [CHMP2b + BIC/4AP], 17 [CHMP2b[intron5]], 13 [CHMP2b[intron5]+BIC/4AP] cells/condition; N ≥ 3 independent experiments). For WT and mutant CHMP2b, quantification only includes PLA puncta colocalized with the mCh-CHMP2b signal. Scale bar = 10 μm. All scatter plots show the mean ± SEM.

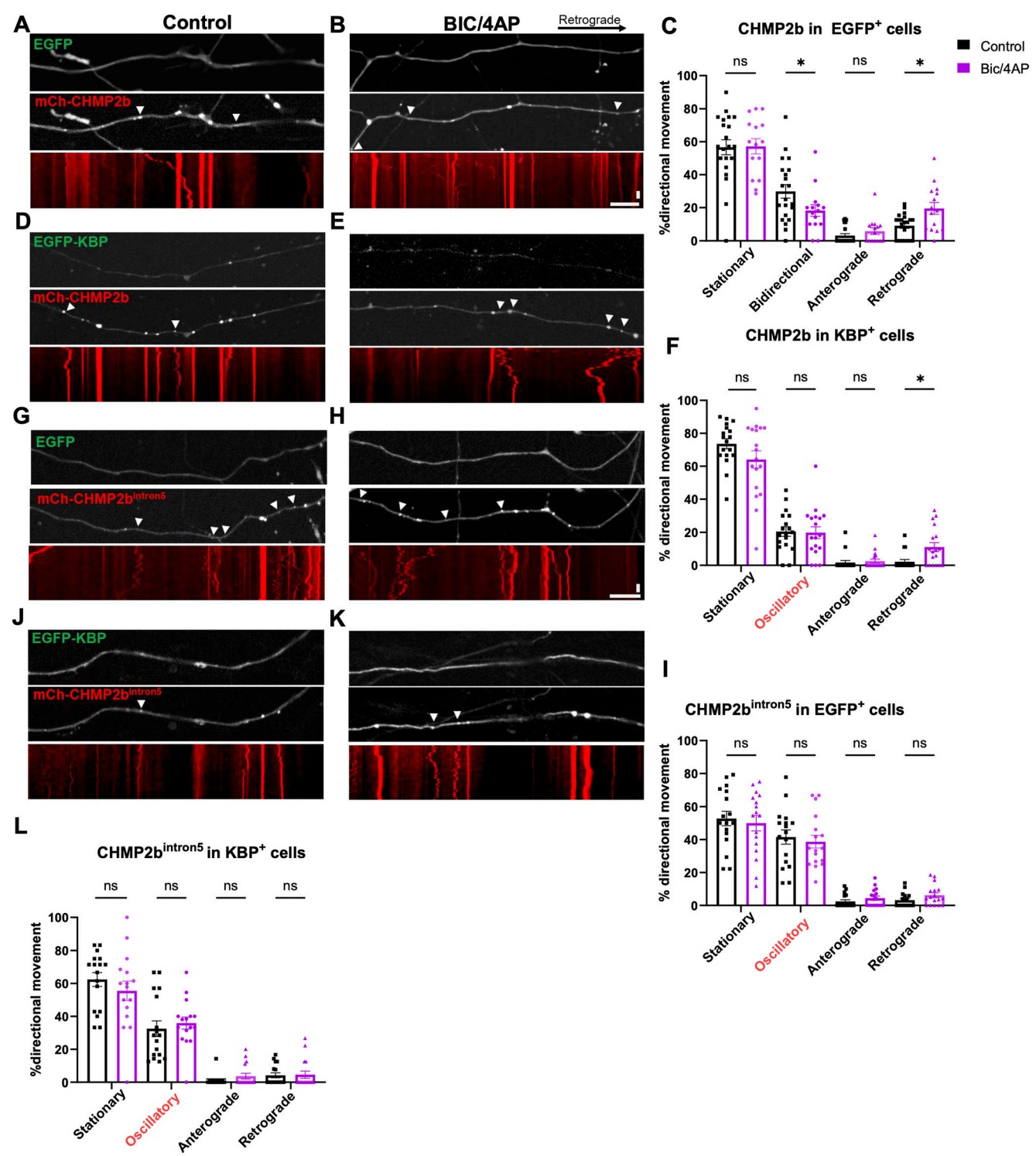

**Figure 8. KBP regulates the transport of CHMP2b but not CHMP2b$^{intron5}$.**

**(A, B)** Still images and associated kymographs of 14 DIV hippocampal axons coexpressing mCh-CHMP2b and EGFP, under control (A) or BIC/4AP conditions (B). **(C)** Percentage of mCh-CHMP2b puncta from stationary, bidirectional, anterograde, or retrograde categories of movement in EGFP$^+$ axons (*$P$ = 0.0154, *$P$ = 0.0327, one-way ANOVA with Dunn's multiple comparisons test; n = 21 [control], 15 [BIC/4AP] axons/condition; N ≥ 3 independent experiments). **(D, E)** Still images and associated kymographs of 14 DIV hippocampal axons coexpressing mCh-CHMP2b and EGFP-KBP, under control (D) or BIC/4AP conditions (E). **(F)** Percentage of mCh-CHMP2b puncta from stationary, "oscillatory," anterograde, or retrograde categories of movement in KBP$^+$ axons (*$P$ = 0.0341, one-way ANOVA with Dunn's multiple comparisons test; n = 18 [control], 18 [BIC/4AP] axons/condition; N ≥ 3 independent experiments). **(G, H)** Still images and associated kymographs of 14 DIV hippocampal axons coexpressing mCh-CHMP2b$^{intron5}$ and EGFP, under control (G) or BIC/4AP conditions (H). **(I)** Percentage of mCh-CHMP2b$^{intron5}$ puncta from stationary, oscillatory, anterograde, or retrograde categories of movement in EGFP$^+$ axons (ns, one-way ANOVA with Dunn's multiple comparisons test; n = 17 [control], 17 [BIC/4AP] axons/condition; N ≥ 3 independent

imaging experiments, neurons were plated onto glass-bottom 50-mm dishes (MatTek; 175,000 cells/dish) or 12-mm coverslips (60,000 cells/coverslip) prepared as previously described (Sheehan et al, 2016; Birdsall et al, 2022). Neuro2a (N2a) neuroblastoma cells (ATCC, CCL-131) and HEK293T cells (Sigma-Aldrich) were cultured in DMEM/GlutaMAX (Thermo Fisher Scientific/Invitrogen) with 10% FBS (Atlanta Biological) and Antibiotic–Antimycotic (Thermo Fisher Scientific/Invitrogen) and kept at 37°C in 5% $CO_2$. For neuronal transduction, lentivirus was produced as previously described (Sheehan et al, 2016; Birdsall et al, 2022) and neurons were transduced on 7 DIV and either imaged or fixed and stained on 13–15 DIV. Alternatively, neurons were transfected on 12–14 DIV using Lipofectamine 2000 (Thermo Fisher Scientific/Invitrogen) following the manufacturer's recommendations and imaged 24–48 h later.

## Plasmids and reagents

pFU-mCh-Rab5a and pFU-mCh-Rab7 were described in Sheehan et al (2016); pFU-mCh-Hrs-W and pFU-EGFP-Chmp4b-W were described in Birdsall et al (2022). For CHMP4b, EGFP was replaced by mCherry at the AgeI/BsrGI sites to create pFU-mCh-CHMP4b-W. Human CHMP2b (NCBI accession# NM_014043.4) and CHMP2b[intron5] (same sequence with M178V mutation followed by a stop codon) were synthesized (Genewiz), subcloned into pEGFP-N2 and pmCherry-N2 plasmids using XhoI/BamHI sites, then subcloned into the FUGWm backbone using XbaI/EcoRI sites (FUGWm) and NheI/MfeI sites (for CHMP2b/CHMP2[intron5] inserts) to generate the final constructs pFU-EGFP/mCh-CHMP2b-W and pFU-EGFP/mCh-CHMP2b[intron5]-W. For the expression of shCtrl or shHrs with EGFP-Synapsin1a, the latter was synthesized (Genewiz) and subcloned into OriGene pGFP-C-shLenti vectors containing shCtrl or shHrs C (catalog #TL710380) using BsrGI/XhoI sites. Human KIAA1279 (KBP; NCBI accession# NM_015634.4) was synthesized (Genewiz) and subcloned into the FUGWm vector using BsrGI/XhoI sites to generate pFU-EGFP-KBP. The previously published shKBP sequence (Brouwers et al, 2017) was synthesized (Genewiz) and subcloned into pFUGW-H1 using EcoRI/PacI sites.

shCtrl sequence: 5′-GCACTACCAGAGCTAACTCAGATAGTACT-3′
shHrs sequence: 5′-CCGCAGTATCACCAATGACTCGGCTGTGC-3′
shKBP sequence: 5′-TATCATAGTAAGCATGTGCTT-3′

## Antibodies and chemical reagents

The following antibodies were used: tubulin mouse antibody (#t9026; Sigma-Aldrich), mCherry rabbit antibody (PA5-34974; Invitrogen), mCherry mouse antibody (ab125096; Abcam), Hrs rabbit antibody (D7T5N; Cell Signaling), GFP mouse antibody (#11814460001; Roche), VAMP2 (104 211; Synaptic Systems), and KBP rabbit antibody (KIAA1279; 25653-1-AP; Proteintech). Pharmacological agents were used in the following concentrations and time courses: bicuculline (40 $\mu$M, 2 h; Sigma-Aldrich), 4-aminopyridine (50 $\mu$M, 2 h; Tocris Bioscience), and LysoTracker Red DND-99 (50 nM, 30 min; Thermo Fisher Scientific). Unless otherwise indicated, all other chemicals were purchased from Sigma-Aldrich.

## Live imaging

Imaging was conducted in Neurobasal medium, minus phenol red (Thermo Fisher Scientific), supplemented with 5% GlutaMAX (Thermo Fisher Scientific) and 10% NeuroCult SM1 (STEMCELL Technologies). Neurons were imaged on a Nikon CSU-W1 SoRa confocal microscope equipped with a Yokogawa spinning disk, a motorized xy stage with Piezo z-drive, SR Plan Apo 60X/1.27 NA objective, LAPP Illuminator, and 2x Photometrics cMOS cameras in a temperature- and $CO_2$-controlled chamber. Images were obtained in NIS-Elements software at 1024 × 1024 resolution. Axons were identified via location and morphological criteria, and imaged ~100–300 $\mu$m away from the soma. One frame was acquired every 4s for 4 min, for a total of 60 frames. Image processing was performed using Fiji/ImageJ. For LysoTracker imaging, the dye was added to fresh Neurobasal media without phenol red using manufacturer-recommended concentrations 30 min before imaging, and washed out 3x with prewarmed media. Cells were returned to their original media for imaging.

## Immunofluorescence microscopy

Neurons were immunostained as described previously (Sheehan et al, 2016; Birdsall et al, 2022), with coverslips mounted in Aqua-Poly/Mount (Polysciences) and dried overnight in the dark. Images were acquired on the Nikon CSU-W1 SoRa confocal microscope, and image processing was performed using Fiji/ImageJ.

## Live-imaging analysis

Motility and directionality analyses were performed as described in Birdsall et al (2022). Only puncta between 0.5 and 1.5 $\mu$m in diameter were analyzed. Briefly, directionality was categorized as follows: total displacement ≥4 $\mu$m away from the cell body = anterograde; ≥4 $\mu$m toward the cell body = retrograde; ≥4 $\mu$m total displacement with <4 $\mu$m in one direction = bidirectional; <1 $\mu$m net displacement = stationary. "Oscillatory" was a term used in experiments involving CHMP2b[intron5] or KBP (Figs 4, 8, and 9) to describe the short bidirectional movements defined as <4 $\mu$m but ≥1 $\mu$m total displacement with <4 $\mu$m displacement in one direction. For directionality and % movement analyses, axons were only analyzed if they had at least 4 puncta and a minimum of 2 with net displacements ≥4 $\mu$m. Graphs of directional movement show the percentage of bidirectional, retrograde, or anterograde puncta over the total number of puncta (both stationary and moving). A value of 0 in these graphs indicates the absence of that particular type of directional movement for a given axon.

experiments). **(J, K)** Still images and associated kymographs of 14 DIV hippocampal axons coexpressing mCh-CHMP2b[intron5] and EGFP-KBP, under control (J) or BIC/4AP conditions (K). **(L)** Percentage of mCh-CHMP2b[intron5] puncta from stationary, "oscillatory," anterograde, or retrograde categories of movement in KBP[+] axons (*$P$ = 0.0341, one-way ANOVA with Dunn's multiple comparisons test; n = 16 [control], 16 [BIC/4AP] axons/condition; N ≥ 3 independent experiments). For all images, horizontal size bar = 10 $\mu$m, vertical scale bar = 30 s. White arrowheads indicate motile puncta. All scatter plots show the mean ± SEM.

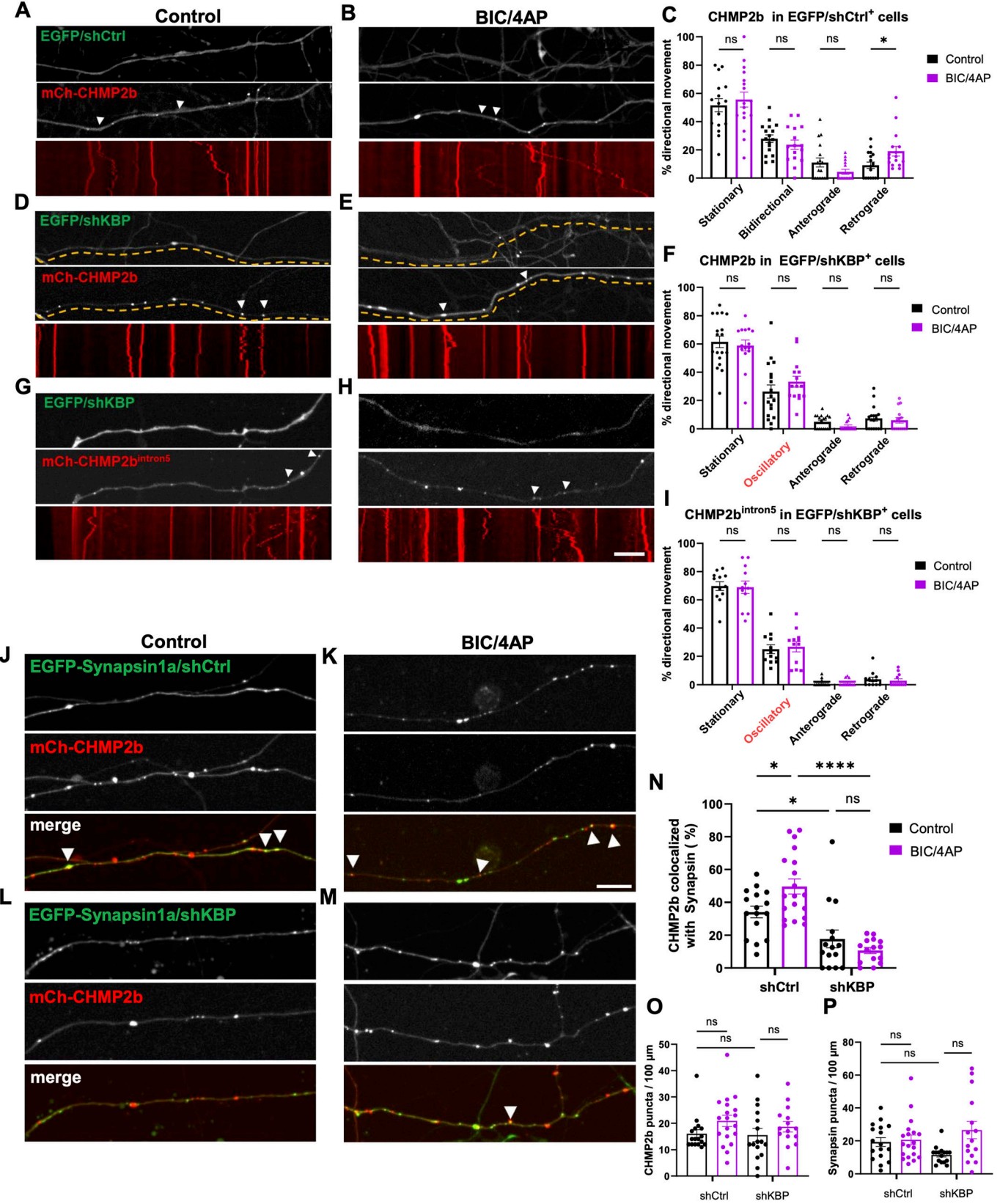

Kymographs were generated using the Velocity Measurement Tool (https://dev.mri.cnrs.fr/projects/imagej-macros/wiki/Velocity_Measurement_Tool). The tracks of each kymograph were traced with the segmented line tool, with displacement and displacement velocity estimated from these traces using the "velocity" function in the Velocity Measurement Tool program.

## Colocalization analysis

ROIs were drawn around the axon or cell body of interest using the "Freehand line" tool in Fiji/ImageJ software. Colocalization analysis was performed semi-automatically using the ComDet v.0.5.5 plugin (Author: Eugene Katrukha). The threshold for detection in this program was determined empirically for each channel (typically 8–20). A colocalization event was defined as one in which the maximum intensity pixels of each channel were no more than three pixels apart from one another (the average size of CHMP2b puncta); thus, partial or total overlap of two puncta is counted as colocalization. The percent colocalization was calculated within the plugin.

Cotransport is defined here as the occurrence of two fluorescence puncta moving together at least 50% of the imaging period (at least 2 min). This is calculated by the fraction of colocalized puncta that move ≥4 $\mu$m divided by the total number of colocalized puncta (i.e., moving colocalized puncta/[moving colocalized puncta + stationary colocalized puncta]). The total number of colocalized puncta was determined using the colocalization analysis procedure described above. Each colocalized punctum was then vetted for processive/nonprocessive movement ≥4 $\mu$m using kymographs generated from each live-imaging video. If net displacement was ≥4 $\mu$m and at least 50% of the kymograph tracks in the GFP and mCherry channels overlapped, then those proteins were considered cotransported.

## Immunoprecipitation and Western blot

For immunoprecipitation assays, N2a cells were transfected at 70% confluence using Lipofectamine 3000 (Thermo Fisher Scientific /Invitrogen) according to the manufacturer's instructions. Cells were collected 48 h later in ice-cold lysis buffer (50 mm Tris base, 150 mm NaCl, 1% Triton X-100, 0.5% deoxycholic acid) with protease inhibitor mixture (Roche) and clarified by centrifugation at high speed (20,000 rcf). The resulting supernatant was incubated with ChromoTek GFP-Trap Agarose beads (Proteintech). Lysates and beads were incubated at 4°C under constant rotation overnight. Beads were washed 2–3× with PBS containing 0.05% Triton (PBST) and then once with PBS. Bound proteins were eluted using 2x SDS sample buffer (Bio-Rad) and subject to SDS–PAGE immunoblotting as described in the following. For other immunoblotting experiments from cultured neurons, lysates were collected directly in 4× SDS sample buffer (Bio-Rad). Samples were subjected to SDS–PAGE, transferred to nitrocellulose membranes, and probed with the primary antibody in 5% BSA/PBS + 0.05% Tween-20 overnight at 4°C, followed by DyLight 680 or 800 anti-rabbit, anti-mouse, or anti-goat secondary antibodies (Thermo Fisher Scientific) for 1 h. Membranes were imaged using Odyssey Infrared Imager (Model 9120; LI-COR Biosciences). The protein intensity was measured using the "Gels" function in Fiji.

## Proximity ligation assay

PLA was performed in hippocampal neurons according to the manufacturer's instructions (Duolink; Sigma-Aldrich). PLA probes were diluted in Duolink blocking solution, and the primary antibody pairs used were anti-KIAA1279 (KBP) (rabbit, 1:100) and anti-mCherry (mouse, 1:500). All protocol steps were performed at 37°C in a humidified chamber, except for the washing steps. Coverslips were then mounted using Duolink In Situ Mounting Media with DAPI and imaged within 1–2 d.

## Statistical analyses

All statistical analyses were performed in GraphPad Prism 9/10. Statistics for each experiment are described in the Figure Legends. A D'Agostino–Pearson normality test was used to assess normality. An unpaired $t$ test or a Mann–Whitney $U$ test was used to compare

---

**Figure 9. KBP knockdown abolishes CHMP2b directional transport.**
**(A, B)** Still images and associated kymographs of 14 DIV hippocampal axons coexpressing mCh-CHMP2b and EGFP/shCtrl, under control (A) or BIC/4AP conditions (B). **(C)** Percentage of mCh-CHMP2b puncta from stationary, bidirectional, anterograde, or retrograde categories of movement in EGFP/shCtrl[+] axons (*P = 0.0429, one-way ANOVA with Dunn's multiple comparisons test; n = 17 [control], 15 [BIC/4AP] axons/condition; N ≥ 3 independent experiments). **(D, E)** Still images and associated kymographs of 14 DIV hippocampal axons coexpressing mCh-CHMP2b and EGFP/shKBP, under control (D) or BIC/4AP conditions (E). **(F)** Percentage of mCh-CHMP2b puncta from stationary, oscillatory, anterograde, or retrograde categories of movement in EGFP/shKBP[+] axons (ns, one-way ANOVA with Dunn's multiple comparisons test; n = 18 [control], 15 [BIC/4AP] axons/condition; N ≥ 3 independent experiments). **(G, H, I)** Still images of 14 DIV hippocampal axons and associated kymographs coexpressing mCh-CHMP2b[intron5] and EGFP/shKBP, under both control (G) and BIC/4AP conditions (H). **(I)** Percentage of mCh-CHMP2b[intron5] puncta from stationary, oscillatory, anterograde, or retrograde categories of movement in EGFP/shKBP[+] axons (ns, one-way ANOVA with Dunn's multiple comparisons test; n = 12 [control], 12 [BIC/4AP] axons/condition; N ≥ 3 independent experiments). **(J, K)** Single-channel and merged images of 14 DIV hippocampal axons expressing mCh-CHMP2b and EGFP-Synapsin together with a control shRNA (shCtrl) and treated for 2 h with vehicle control (J) or BIC/4AP (K). **(L, M)** Single-channel and merged images of 14 DIV hippocampal axons expressing mCh-CHMP2b and EGFP-Synapsin together with an shRNA to knock down KBP (shKBP), under control (L) or BIC/4AP (M) conditions. **(N)** Percentage of mCh-CHMP2b puncta that colocalize with EGFP-Synapsin (*P = 0.0413, *P = 0.0413, ****P < 0.0001, one-way ANOVA with Tukey's multiple comparisons test; n = 16 [shCtrl], 19 [shCtrl + BIC/4AP], 15 [shKBP], 15 [shKBP + BIC/4AP] axons/condition; N ≥ 3 independent experiments). **(O)** Average number of mCh-CHMP2b puncta per 100 $\mu$m of axon (ns, one-way ANOVA with Tukey's multiple comparisons test; n = 17 [shCtrl], 17 [shCtrl + BIC/4AP], 16 [shKBP], 16 [shKBP + BIC/4AP] axons/condition; N ≥ 3 independent experiments). **(P)** Average number of EGFP-Synapsin puncta per 100 $\mu$m of axon (ns; one-way ANOVA with Tukey's multiple comparisons test; n = 17 [shCtrl], 19 [shCtrl + BIC/4AP], 16 [shKBP], 15 [shKBP + BIC/4AP] axons/condition; N ≥ 3 independent experiments). For all images, horizontal size bar = 10 $\mu$m, vertical scale bar = 30 s. White arrowheads indicate motile puncta (A, B, C, D, E, F, G, H) or CHMP2b colocalized with Synapsin (J, K, L, M). Yellow dashed lines outline the axon. All scatter plots show the mean ± SEM.

two datasets, and an ANOVA or a Kruskal–Wallis test was used to compare three or more datasets where appropriate. For multiple comparisons, Tukey's or Dunn's post hoc test was used to compare each mean with one another. *n* indicates the number of events or cells pooled across at least three trials per experiment. *N* indicates the number of biological replicates. Bars represent the mean ± SEM. Statistical significance is noted as follows (and in Figure Legends): ns, $P > 0.05$; $*P \leq 0.05$; $**P \leq 0.01$; $***P \leq 0.001$, $****P \leq 0.0001$.

# Data Availability

The data that support the findings of this study are available from the corresponding author upon reasonable request.

# Supplementary Information

# Acknowledgements

This work was supported by NIH grants 2R01NS080967 and 1RF1AG069941 to CL Waites, NSF grant DGE-2036197 to KR Kirwan, and a TIGER (Taub Institute Grants for Emerging Research) Award to CL Waites.

## Author Contributions

KR Kirwan: conceptualization, methodology, investigation, formal analysis, data curation, visualization, and writing—original draft, review, and editing.
V Puerta-Alvarado: conceptualization, methodology, funding acquisition, project administration, supervision, and writing—original draft, review, and editing.
CL Waites: investigation and validation.

## Conflict of Interest Statement

The authors declare that they have no conflict of interest.

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
