## [Reviewer comments · Life Science Alliance]

Life Science Alliance

Axonal transport of CHMP2b is regulated by kinesin binding protein and disrupted by CHMP2bintron5

Konner Kirwan, Veria Puerta-Alvarado, and Clarissa Waites

DOI: <https://doi.org/10.26508/lsa.202402934>

Corresponding author(s): Clarissa Waites, Columbia University Irving Medical Center

Review Timeline:

Submission Date:	2024-07-09
Editorial Decision:	2024-09-10
Revision Received:	2025-01-12
Editorial Decision:	2025-02-05
Revision Received:	2025-02-13
Accepted:	2025-02-14

Transaction Report:

September 10, 2024

Re: Life Science Alliance manuscript #LSA-2024-02934-T

Dr. Clarissa Waites
Columbia University Medical Center
Columbia University College of Physicians and Surgeons Dept of Pathology and Cell Biology
630 W. 168th St.
Black Building 1212
New York, NY 10032

Dear Dr. Waites,

Thank you for submitting your manuscript entitled "Axonal transport of CHMP2b is disrupted by the FTD-associated CHMP2bintron5 mutation" to Life Science Alliance. The manuscript was assessed by expert reviewers, whose comments are appended to this letter. We invite you to submit a revised manuscript addressing the Reviewer comments.

Thank you for this interesting contribution to Life Science Alliance. We are looking forward to receiving your revised manuscript.

Sincerely,

B. MANUSCRIPT ORGANIZATION AND FORMATTING:

Reviewer #1 (Comments to the Authors (Required)):

This study examines axon transport of an ESCRT protein CHMP2B and how an FTD-linked mutation in CHMP2B alters its axonal trafficking. The authors show that exogenously expressed CHMP2B displays altered axonal transport with increased neuronal activity and exhibits increased co transport/localization with a presynaptic protein under these conditions. Lastly, they identify a kinesin binding protein as a potential partner for CHMP2B and propose that this interaction is altered with the CHMP2B mutation and potentially reduces the retrograde movement of CHMP2B vesicles by altering the kinesin inhibition that this interaction would bring about. The study presents interesting new findings in areas that are of strong interest to the cell biology and neuroscience community. Examination of the ESCRT proteins, especially in the context of axon transport is relatively under explored, so this work is important.

However, there are some concerns that need to be addressed prior to publication.

1. While the data of co-transport of CHMP2B with Synaptophysin upon altering neuronal activity is interesting, the statement that its transport and recruitment to presynaptic boutons seems to be a bit of a stretch, based on just this data. Tempering the statement to reflect co transport and clarifying that is based on exogenously expressed proteins would be helpful.
2. Further strengthening this claim by examining how the transport is changes on reducing neuronal activity/local and acute activity change such as uncaging may help support the finding. Likewise, examining other presynaptic cargo would also be supportive.
3. While the data on transport is interesting, the use of only exogenously expressed constructs does raise some concerns; it is known that highly expressed proteins in axons can aggregate and end up in autophagosomes and would essentially amount to monitoring autophagosomes rather than a physiologically relevant protein trafficking event. These would in fact show the lysotracker positivity and predominant retrograde motility that is observed with the CHMP2B. While endogenous tagging and examining could be beyond the scope of this study, the increased colocalization of CHMP2B with presynaptic boutons on altering neuronal activity could easily be monitored using immunostaining for endogenous proteins; good antibodies to both proteins exist in the field.
4. Minor concerns: The images of the PLA in figure 7C, D appear a bit blown out. Likewise, the IP in 7A is not very convincing-given the mCherry control lanes.

Reviewer #2 (Comments to the Authors (Required)):

In the paper "Axonal transport of CHMP2b is disrupted by the FTD-associated CHMP2bintron5 mutation", the authors suggest an interesting new mechanism for the regulation of axonal transport of CHMP2b+ vesicles. They also show that wild-type CHMP2b localises to synapses and that both Hrs levels and neuronal activity modulate CHMP2b synaptic recruitment. On the other hand, the FTD-causing mutant CHMP2bintron5, a truncated form of CHMP2b, shows impaired recruitment to synapses and its synaptic localisation is insensitive to neuronal activity and Hrs levels modulation. Moreover, CHMP2bintron5+ vesicles have defective maturation and altered axonal transport.

The authors provide an extensive characterisation of the motile properties of wild-type and mutant CHMP2b-containing vesicles and describe how they are affected by neuronal activity. They show that CHMP2bintron5+ puncta engage less in processive directional movements and that their motility is insensitive to neuronal activity. Reduction of KBP (Kinesin Binding Protein) levels mimics the effects of CHMP2bintron5, affecting the transport of wild-type CHMP2b and abolishing the increase in its retrograde transport observed upon neuronal stimulation. Interestingly, while KBP interacts with CHMP2b, its association with CHMP2bintron5 is impaired. The role of KBP in the regulation of CHMP2b transport in response to neuronal activity, as well as the impairment of this mechanism by CHMP2bintron5, are significant findings that provide advances in the understanding of the trafficking of these vesicles along axons.

However, further experiments are needed to clarify whether the impairment in CHMP2bintron5 axonal transport is mechanistically linked to its reduced synaptic localisation and to the defective maturation of CHMP2bintron5+ vesicles.

Main points

The main finding is the alteration of CHMP2bintron5 axonal transport and the role of the KBP-CHMP2b interaction in its regulation. This is overall well supported, but some experiments and additional analysis are required.

The interaction of KBP with CHMP2b and CHMP2bintron5 is tested using two independent techniques and this is needed given its importance for the proposed mechanism. However, the immunoprecipitation reported in Fig. 7A is not convincing, as there is a significant amount of EGFP-KBP also in the IP lane of the negative control (mCherry only). I strongly recommend optimising this pull-down assay as well as performing the reverse immunoprecipitation (pull down EGFP-KBP and probe for interactions). The PLA results support the conclusion of the authors. However, it is good practice to include a control to ensure the specificity of the signal. In this case, please add a condition where the PLA is performed also in cells expressing mCherry alone. Another key point in the paper is that CHMP2bintron5 fails to localise to the synapse and that its expression affects vesicles maturation. While these are interesting observations, the authors suggest a causal link between these events and the impaired interaction between KBP and CHMP2bintron5 (discussion, page 12) which does not find enough support in the experimental evidence provided. An assessment of CHMP2b synaptic localisation and CHMP2b+ vesicles maturation in KBP-depleted cells would help clarifying this point. Otherwise, the authors should discuss the alternative mechanisms through which CHMP2bintron5 might alter its synaptic localisation and vesicle maturation, independently from the impairment of its axonal transport. The authors propose a dominant negative effect of CHMP2bintron5 on WT CHMP2b transport, but it is unclear whether this applies also to its synaptic localisation. I suggest an experiment where the recruitment of WT-CHMP2b to synapses is assessed in presence of CHMP2bintron5.

Other points:

- Figure 1. Please add quantification for the number of CHMP2b+ puncta in the axons (like for Synapsin in 1F). This is important to clarify whether Hrs affects CHMP2b synaptic recruitment or its axonal localisation instead.
- Figure 2C. It is not clear from the figure legend that the percentage refers to the number of puncta assigned to a specific movement category. I suggest renaming the y axis.
- Figure 2C. I suggest including the percentage of stationary puncta. The data in 2C together with 2D indicate that the increase in the percentage of retrogradely transported puncta comes at the expense of the percentage of stationary ones (rather than a switch from bidirectional to retrograde transport), but this should be made clearer.
- Figure 4D. Like for Figure 2, please include the percentage of stationary puncta. Does CHMP2bintron5 affect the percentage of motile puncta or only their motile properties (i.e. oscillatory vs bidirectional movements)? I suggest adding the percentage of motility also for CHMP2bintron5+ puncta (like in 2D for CHMP2b+ puncta). How do they compare?
- Brief movements are called "Oscillatory" also in Figure 4D while in the methods the authors report that this terminology only applies to KBP overexpression/knockdown experiments. A portion of organelles is classified as oscillatory also in 4F (including the WT CHMP2b condition, although this type of movement was not included in the previous analysis of puncta motility in Figure 2). Moreover, the parameters used to discriminate between stationary and oscillatory puncta are not reported. Please clarify how the transport categories are defined univocally and consistently among different experiments.
- Figure 6. Like for Figure 1, please add quantification for the number of CHMP2bintron5+ puncta in the axons. How does this compare to the density of CHMP2b+ puncta? Altered axonal localisation of CHMP2bintron5 might underlie (or contribute to) the reduction in its synaptic levels.
- Figure 8. Please add the data relative to the stationary puncta.
- Although the shRNA used for the knockdown of KBP has been previously reported in published work, it is important to validate that it works reliably in this experimental setting. Please show a western blot validating efficient knockdown of KBP in this system. Moreover, knowing the extent of the KBP depletion sufficient to cause an effect on CHMP2b transport is an information that adds significant value to the understanding of the mechanism described.
- Figure S1B. While the reduction is significant from the quantification, it is not very convincing from the blot. Please also add dots corresponding to each replicate in the bar chart.
- Page 14: "interaction is absent". Please change the wording, the data only show that the interaction is reduced, not absent.
- The paper would benefit from a discussion on the role of Hrs in regulating the synaptic localisation of CHMP2b, but not CHMP2bintron5.

Considering the experiments and changes suggested above, I advise to allow up to 3 months for the resubmission of the manuscript.

Reviewer #3 (Comments to the Authors (Required)):

Kirwan and Waites present a manuscript describing synaptic transport of ESCRT-III-positive structures in rat neuronal cultures. Focussing primarily on CHMP2b, and the FTD-causing C-terminal deletion of CHMP2b (CHMP2bintron5), they observe axonal transport of this protein with a slight stimulus-dependent bias toward retrograde transport of CHMP2b-positive structures, and note that the Intron5 deletion displays less axonal transport and more oscillatory behaviour, which they suggest is due to impaired binding to a kinesin adaptor, KBP.

KBP is a plus end kinesin binding protein that disrupts kinesin MT attachment and thus impairs transport. The authors hypothesise that if Intron5 doesn't interact with KBP, then it will enhance the interaction with other kinesins and that this would somehow lead to oscillatory behaviour of the GFP-/mCh-positive CHMP2bintron5. It isn't clear why this would be the case, and if the author's predictions are correct, it isn't demonstrated that Intron5 (or Intron5-positive structures) binds more kinesins that

wildtype CHMP2b. As such, the mechanistic understanding of this oscillatory behaviour is lacking.

The data is generally clear and the figures are interpretable. I'm struck from the kymographs that the majority of punctate structures observed are static and I think the authors should note that the N-terminal tag on their ESCRT-III subunits might be disrupting the function of these proteins. As such, it isn't really clear what the authors are reporting with these bright puncta.

The interaction data with KBP is not convincing. The pull down assays are bringing down non-specific proteins (e.g., see the mCh co precipitated in lane 2) and the levels of KBP co-precipitated are really very similar between mCh, Wildtype and intron5. If the authors want to propose KBP as a new interaction for CHMP2b, then this area of the manuscript needs to be clearer and less ambiguous. As well as improving the protein-protein interaction, it would help to visualise KBP and investigate it's colocalization with CHMP2b. In the PLA data, the majority of green PLA dots are not congruent with the mCh-CHMP2b dots, so it isn't clear what this assay is reporting.

There are some overinterpretations; notably, the abstract claims CHMP2b transport and recruitment to presynaptic boutons is controlled by stimulus, but I think the authors have only shown that transport is controlled in this way - there's no examination of co-recruitment of CHMP to synapses, rather they show in the first figures that synapsin and ESCRT-positive structures move together. Also, the HRS depletion in S1 is very weak, which makes me question why infer anything about vesicle behaviour when you know the siRNA is barely depleting the target.

In general, I'm not sure what we are learning from this study, beyond observational data that the intron5 mutant causes the CHMP2b positive structures to move less directionally; it is already known that expression of this mutant induces aberrant endosomes, so is somewhat unsurprising that they don't move that much. The relevance of this observation to disease or to neuronal function is absent.

Response to Reviewers

We thank the reviewers for their detailed and thoughtful comments, which have helped us to significantly improve the manuscript. Our responses to their comments and suggestions are below in red.

REVIEWER 1

1. While the data of co-transport of CHMP2B with Synaptophysin upon altering neuronal activity is interesting, the statement that its transport and recruitment to presynaptic boutons seems to be a bit of a stretch, based on just this data. Tempering the statement to reflect co transport and clarifying that is based on exogenously expressed proteins would be helpful.

We understand the reviewer's concern, and would like to clarify that this experiment does not investigate cotransport of CHMP2b with Synapsin1a, but rather the recruitment of CHMP2b to existing presynaptic boutons. Indeed, Synapsin1a is a particularly good marker for presynaptic boutons, as it exhibits very limited punctate/vesicular transport in axons and predominantly accumulates at presynaptic vesicle clusters (Tang et al., *J Neurosci*, 2013). We have changed the wording on page 5 to emphasize this point.

2. Further strengthening this claim by examining how the transport is changes on reducing neuronal activity/local and acute activity change such as uncaging may help support the finding. Likewise, examining other presynaptic cargo would also be supportive.

We further strengthen our claim by showing that tetrodotoxin (TTX), a sodium channel blocker that prevents action potential firing, significantly reduces CHMP2b motility in axons (Figure S1F-H). As discussed above, we also clarify that EGFP-Synapsin1a puncta mark presynaptic sites, and that our CHMP2b/Synapsin1a experiments demonstrate presynaptic localization rather than cotransport with presynaptic cargo (p. 5).

3. While the data on transport is interesting, the use of only exogenously expressed constructs does raise some concerns...While endogenous tagging and examining could be beyond the scope of this study, the increased colocalization of CHMP2B with presynaptic boutons on altering neuronal activity could easily be monitored using immunostaining for endogenous proteins; good antibodies to both proteins exist in the field.

We appreciate and share the reviewer's concern that protein overexpression can induce experimental artifacts. As suggested, we have performed immunostaining experiments with antibodies against endogenous CHMP2b, VAMP2 (another SV-associated presynaptic marker), and neurofilament (an axonal marker) under control and activity conditions to corroborate our overexpression findings. These data, which demonstrate that neuronal activity increases the localization of endogenous CHMP2b to presynaptic boutons (based on its colocalization with VAMP2), are shown in Figure S1A-C.

4. Minor concerns: The images of the PLA in figure 7C, D appear a bit blown out. Likewise, the IP in 7A is not very convincing- given the mCherry control lanes.

We have decreased the brightness of PLA images in Figure 7C-D -- and please note that these images are quantified based on puncta density, not intensity. In addition, we have now optimized the IP experiments using GFP-Trap Agarose (ChromoTek) to pull down EGFP-KBP. Our new results show a strong pulldown of WT CHMP2b, but not CHMP2b^{intron5} or mCherry control (Figure 7A-B).

REVIEWER 2

The interaction of KBP with CHMP2b and CHMP2bintron5 is tested using two independent techniques and this is needed given its importance for the proposed mechanism. However, the immunoprecipitation reported in Fig. 7A is not convincing, as there is a significant amount of EGFP-KBP also in the IP lane of the negative control (mCherry only). I strongly recommend optimising this pull-down assay as well as performing the reverse immunoprecipitation (pull down EGFP-KBP and probe for interactions).

We thank the reviewer for this very helpful suggestion. As described above in our response to Reviewer 1, we have now optimized the IP assay using GFP-Trap Agarose (ChromoTek) to pull down EGFP-KBP, followed by immunoblotting for mCh-tagged CHMP2b and CHMP2b^{intron5} with mCherry antibodies. Our new results show enrichment of GFP in all IP conditions, and notably no pulldown of mCherry in the control lane. Most importantly, we see a robust IP of mCherry-CHMP2b with a significantly weaker IP of mCherry-CHMP2b^{intron5} (Figure 7A-B).

The PLA results support the conclusion of the authors. However, it is good practice to include a control to ensure the specificity of the signal. In this case, please add a condition where the PLA is performed also in cells expressing mCherry alone.

In Figure 7E, we now include the mCherry control condition to confirm specificity of the PLA signal.

Another key point in the paper is that CHMP2bintron5 fails to localise to the synapse and that its expression affects vesicles maturation. While these are interesting observations, the authors suggest a causal link between these events and the impaired interaction between KBP and CHMP2bintron5 (discussion, page 12) which does not find enough support in the experimental evidence provided. An assessment of CHMP2b synaptic localisation and CHMP2b+ vesicles maturation in KBP-depleted cells would help clarifying this point. Otherwise, the authors should discuss the alternative mechanisms through which CHMP2bintron5 might alter its synaptic localisation and vesicle maturation, independently from the impairment of its axonal transport. The authors propose a dominant negative effect of CHMP2bintron5 on WT CHMP2b transport, but it is unclear whether this applies also to its synaptic localisation. I suggest an experiment where the recruitment of WT-CHMP2b to synapses is assessed in presence of CHMP2bintron5.

We thank the reviewer for these insightful comments and suggestions. We have added data showing that knockdown of KBP impairs the synaptic recruitment of CHMP2b under control and activity conditions (Figure 9). We also show that overexpression of CHMP2b^{intron5} inhibits the synaptic localization of CHMP2b (Figure 6G-K).

Other points:

- Figure 1. Please add quantification for the number of CHMP2b+ puncta in the axons (like for Synapsin in 1F). This is important to clarify whether Hrs affects CHMP2b synaptic recruitment or its axonal localisation instead.

We have added a graph showing the number of CHMP2b puncta/100 μm in axons under control and Hrs knockdown conditions (Figure 1G). These data indicate that Hrs affects the synaptic recruitment of CHMP2b, but not its axonal localization.

- Figure 2C. It is not clear from the figure legend that the percentage refers to the number of puncta assigned to a specific movement category. I suggest renaming the y axis.

We have renamed the y axis on this graph '% directional movement'.

- Figure 2C. I suggest including the percentage of stationary puncta. The data in 2C together with 2D indicate that the increase in the percentage of retrogradely transported puncta comes at the expense of the percentage of stationary ones (rather than a switch from bidirectional to retrograde transport), but this should be made clearer.

We have now added the quantification for stationary puncta to this graph, which is unchanged between control and activity conditions. These data indicate that there is indeed a switch from bidirectional and/or anterograde to retrograde transport, although the bidirectional and anterograde categories exhibit non-significant (but trending) changes in response to neuronal activity in Figure 2C. However, please note that there is a significant decrease in CHMP2b bidirectional transport under the neuronal activity (BIC/4AP) condition in Figure 8C, indicating that this effect is subtle and not always captured in our experiments.

- Figure 4D. Like for Figure 2, please include the percentage of stationary puncta. Does CHMP2b^{intron5} affect the percentage of motile puncta or only their motile properties (i.e. oscillatory vs bidirectional movements)? I suggest adding the percentage of motility also for CHMP2b^{intron5+} puncta (like in 2D for CHMP2b+ puncta). How do they compare?

We now include stationary puncta in Figure 4D, and the percent motility of CHMP2b^{intron5} (~40%) in Figure 4E. This latter value is higher than the baseline motility of WT CHMP2b (~25%; Figure 2D), suggesting that the intron5 mutation causes an increase in the percentage of motile CHMP2b puncta. However, note that in contrast to WT CHMP2b, the motility of CHMP2b^{intron5} is not increased by neuronal activity.

- Brief movements are called "Oscillatory" also in Figure 4D while in the methods the authors report that this terminology only applies to KBP overexpression/knockdown experiments. A portion of organelles is classified as oscillatory also in 4F (including the WT CHMP2b condition, although this type of movement was not included in the previous analysis of puncta motility in Figure 2). Moreover, the parameters used to discriminate between stationary and oscillatory puncta are not reported. Please clarify how the transport categories are defined univocally and consistently among different experiments.

We apologize for the confusion created by our use of the 'oscillatory' term. We have stated the parameters used to distinguish oscillatory from bidirectional puncta in the Results section (p. 8) and those used distinguish stationary from oscillatory puncta in the Methods section under 'Live imaging analysis' (p. 19).

- Figure 6. Like for Figure 1, please add quantification for the number of CHMP2b^{intron5+} puncta in the axons. How does this compare to the density of CHMP2b+ puncta? Altered axonal localisation of CHMP2b^{intron5} might underlie (or contribute to) the reduction in its synaptic levels.

We have added a graph showing the number of CHMP2b^{intron5} puncta/100 μ m in axons (Figure 6F). The density of CHMP2b puncta in axons is similar for wild type and mutant protein, suggesting that the mechanisms responsible for CHMP2b's synaptic localization may be distinct from those responsible for its axonal localization.

- Figure 8. Please add the data relative to the stationary puncta.

We have added the quantification for stationary puncta to these graphs.

- Although the shRNA used for the knockdown of KBP has been previously reported in published work, it is important to validate that it works reliably in this experimental setting. Please show a western blot validating efficient knockdown of KBP in this system. Moreover, knowing the extent of the KBP depletion sufficient to cause an effect on CHMP2b transport is an information that adds significant value to the understanding of the mechanism described.

We agree with this point, and the data has been added in Figure S2D-E.

- Figure S1B. While the reduction is significant from the quantification, it is not very convincing from the blot. Please also add dots corresponding to each replicate in the bar chart.

We now provide a more representative blot and have added the individual replicates to the bar graph (Fig. S1D, E).

- Page 14: "interaction is absent". Please change the wording, the data only show that the interaction is reduced, not absent.

We have changed the wording to reflect this.

- The paper would benefit from a discussion on the role of Hrs in regulating the synaptic localisation of CHMP2b, but not CHMP2bintron5.

We have added discussion of this topic to the Discussion section (pp. 13-14).

REVIEWER 3

KBP is a plus end kinesin binding protein that disrupts kinesin MT attachment and thus impairs transport. The authors hypothesize that if Intron5 doesn't interact with KBP, then it will enhance the interaction with other kinesins and that this would somehow lead to oscillatory behaviour of the GFP-/mCh-positive CHMP2bIntron5. It isn't clear why this would be the case, and if the author's predictions are correct, it isn't demonstrated that Intron5 (or Intron5-positive structures) binds more kinesins than wildtype CHMP2b. As such, the mechanistic understanding of this oscillatory behaviour is lacking.

Our working hypothesis (shown in Figure 10) is that KBP binding to CHMP2b inhibits its kinesin-directed transport, leading to more dynein-directed retrograde transport. In contrast, the inability of CHMP2b^{intron5} to interact with KBP is permissive for kinesin-mediated transport and leads to tug-of-wars between kinesins and dynein, resulting in oscillatory behavior. We have tested this hypothesis through functional studies (i.e., imaging of CHMP2b axonal transport under KBP gain- and loss-of-function) rather than biochemical studies. We acknowledge that these experiments do not provide a specific mechanistic understanding of which kinesins regulate CHMP2b transport and oscillatory behavior. However, given the large amount of data already included in the current study, we think it is more appropriate to address this important question in a future study. KBP has been shown to interact with eight kinesins (Kevanaar et al., 2016; Malaby et al., 2016), which will serve as a good starting point for our follow-up investigations.

The data is generally clear and the figures are interpretable. I'm struck from the kymographs that the majority of punctate structures observed are static and I think the authors should note that the N-terminal tag on their ESCRT-III subunits might be disrupting the function of these proteins. As such, it isn't really clear what the authors are reporting with these bright puncta.

We acknowledge that the majority of CHMP2b puncta are stationary, consistent with other studies of ESCRT transport dynamics (Clayton et al., 2018; Birdsall et al., 2022). We have added stationary puncta to all graphs to be transparent about these measurements. Please note that we exclude puncta larger than 7 pixels (the average size of CHMP2b puncta) from our analyses, as many of these are static and could potentially be artifacts of overexpression. Moreover, as discussed above in our response to Reviewer 2, we have corroborated some of our overexpression findings (e.g., the activity-dependent recruitment of Hrs and CHMP2b to presynaptic boutons) with the endogenous proteins by immunostaining.

The interaction data with KBP is not convincing. The pull down assays are bringing down non-specific proteins (e.g., see the mCh co precipitated in lane 2) and the levels of KBP co-precipitated are really very similar between mCh, wildtype and intron5. If the authors want to propose KBP as a new

interaction for CHMP2b, then this area of the manuscript needs to be clearer and less ambiguous. As well as improving the protein-protein interaction, it would help to visualise KBP and investigate its colocalization with CHMP2b.

The other reviewers share this concern. As discussed above, we now include new data from optimized IP experiments (Figure 7A-B), showing negligible pulldown of mCh and mCh-CHMP2b^{intron5} but robust pulldown of WT mCh-CHMP2b.

In the PLA data, the majority of green PLA dots are not congruent with the mCh-CHMP2b dots, so it isn't clear what this assay is reporting.

There are indeed examples of PLA puncta that do not colocalize with overexpressed CHMP2b. We are not sure what these represent, as we see very little nonspecific PLA signal in the absence of mCh-CHMP2b expression (Figure 7E). In our analysis, we only quantify PLA reactions that colocalize with CHMP2b signal (the majority of the signal) and we have clarified this in the figure legend.

There are some overinterpretations; notably, the abstract claims CHMP2b transport and recruitment to presynaptic boutons is controlled by stimulus, but I think the authors have only shown that transport is controlled in this way - there's no examination of co-recruitment of CHMP to synapses, rather they show in the first figures that synapsin and ESCRT-positive structures move together.

As discussed above in our response to Reviewer 1, these experiments do not depict cotransport of CHMP2b with Synapsin. Indeed, Synapsin was chosen for these studies precisely because it does not undergo vesicle-based axonal transport and is predominantly associated with SV pools at presynaptic boutons. We have now clarified this point in the text (p. 5).

Also, the HRS depletion in S1 is very weak, which makes me question why infer anything about vesicle behaviour when you know the siRNA is barely depleting the target.

As discussed above in our response to Reviewer 2, we now provide a better representative blot for Hrs knockdown and have added individual replicates quantifying this knockdown to the bar graph.

In general, I'm not sure what we are learning from this study, beyond observational data that the intron5 mutant causes the CHMP2b positive structures to move less directionally; it is already known that expression of this mutant induces aberrant endosomes, so is somewhat unsurprising that they don't move that much. The relevance of this observation to disease or to neuronal function is absent.

We thank the reviewer for these comments, as they have helped us to better articulate and communicate the significance of this study. The key contributions of this work are as follows: 1) we provide the first characterization of the axonal transport of ESCRT-III component CHMP2b, which is critical for MVB formation and commonly used as a late endosome marker. This study clarifies how CHMP2b trafficking is similar and different from other ESCRT (i.e., Hrs) and late endosome (i.e., LAMP1, Rab7) proteins; 2) we describe a novel mechanism by which CHMP2b axonal transport and presynaptic localization is regulated through an interaction with KBP; and 3) we provide evidence that this interaction is impaired by the FTD-causative CHMP2b^{intron5} mutation, likely contributing to the presynaptic defects observed in carriers of this mutation (mouse models and, in all probability, humans). We acknowledge that our findings are more relevant for those who study neuronal cell biology and axonal trafficking mechanisms versus neurodegenerative disease etiology, and we have carefully edited the manuscript so as not to oversell the disease relevance of our findings.

February 5, 2025

RE: Life Science Alliance Manuscript #LSA-2024-02934-TR

Dr. Clarissa Waites
Columbia University Irving Medical Center
Columbia University College of Physicians and Surgeons Dept of Pathology and Cell Biology
630 W. 168th St.
Black Building 1212
New York, NY 10032

Dear Dr. Waites,

Thank you for submitting your revised manuscript entitled "Axonal transport of CHMP2b is regulated by kinesin binding protein and disrupted by CHMP2bintron5". We would be happy to publish your paper in Life Science Alliance pending final revisions necessary to meet our formatting guidelines.

- please address the Reviewers' remaining points
- please be sure that the authorship listing and order is correct
- please upload all figure files as individual ones, including the supplementary figure files; all figure legends should only appear in the main manuscript file
- please add the Twitter and Bluesky handles of your host institute/organization as well as your own or/and one of the authors in our system
- please rename Bibliography to References
- please add your main, supplementary figure, and video legends to the main manuscript text after the references section
- there is a callout for figure S1 on pg.7 probably, but the number of the figure is not written...please correct
- please add a callout for Figure S2C to your main manuscript text
- you may want to consider uploading Figure 10 as a Graphical Abstract rather than as a figure, but this it up to you

LSA now encourages authors to provide a 30-60 second video where the study is briefly explained. We will use these videos on social media to promote the published paper and the presenting author (for examples, see <https://docs.google.com/document/d/1-UWCfbE4pGcDdcgzcmiuJl2XMBJnxKYeqRvLLrLS08s/edit?usp=sharing>). Corresponding or first-authors are welcome to submit the video. Please submit only one video per manuscript. The video can be emailed to contact@life-science-alliance.org

A. FINAL FILES:

B. MANUSCRIPT ORGANIZATION AND FORMATTING:

Sincerely,

Reviewer #1 (Comments to the Authors (Required)):

This revised version, in my opinion, addresses the major concerns raised by the reviewers. This includes endogenous staining to support localization data, cleaner IP to support the interaction of CHMP2B and KBP, experiments to support a dominant negative effect of CHMP2bintron5 on transport of WT protein. This is suitable for publication in LSA.

Reviewer #2 (Comments to the Authors (Required)):

In this study, the authors characterise the axonal trafficking of CHMP2b+ vesicles and CHMP2b synaptic recruitment, reporting their regulation by neuronal activity. They also show that KBP plays a pivotal role in the regulation of CHMP2b+ vesicles trafficking and CHMP2b localisation to synapses. Interestingly, the FTD-causing mutant CHMP2bintron5 shows altered axonal trafficking, reduced synaptic recruitment and impaired interaction with KBP. The authors suggest that the presence of CHMP2bintron5 has a dominant effect on CHMP2b+ vesicle trafficking and synaptic localisation. Overall, the authors have addressed the points previously raised and the manuscript is now ready for publication.

Please find below some minor points that I suggest addressing.

- FigS1E. I suggest using a normalisation method that takes into account the variability among biological replicates within the control condition as well.
- Line 179: Typo in the reference to the supplementary figure: missing number. S1F-H
- Fig. 6K: Please double check the colour legend. Have the colours corresponding to control and BIC/4AP of "CHMP2b" been swapped in the graph?
- Fig. 8L: Based on comparison with the previous version of the paper, it looks like there has been a mislabelling on the x axis after the addition of the data relative to the stationary pool. Please double check.
- Fig. S2E: As in fig. S1E, I suggest using a normalisation method that takes into account also the biological variability within the control condition.

Reviewer #3 (Comments to the Authors (Required)):

In the revised manuscript, new data has been added to address many of my points.

However, I still find the interaction data troubling; if KBP binds to the C-terminus of CHMP2B, then it should not be bound by the Intron5 mutant. This isn't the case; from the IP data, the KBP interaction is only reduced. The authors overinterpret this (e.g., L332 - Intron5 does not bind to KBP) and at the very least, these conclusions need to be turned down. Demonstration of endogenous binding would be preferable.

February 14, 2025

RE: Life Science Alliance Manuscript #LSA-2024-02934-TRR

Dr. Clarissa L Waites
Columbia University Irving Medical Center
Columbia University College of Physicians and Surgeons Dept of Pathology and Cell Biology
630 W. 168th St.
Black Building 1212
New York, NY 10032

Dear Dr. Waites,

Thank you for submitting your Research Article entitled "Axonal transport of CHMP2b is regulated by kinesin binding protein and disrupted by CHMP2bintron5". It is a pleasure to let you know that your manuscript is now accepted for publication in Life Science Alliance. Congratulations on this interesting work.

DISTRIBUTION OF MATERIALS:

Again, congratulations on a very nice paper. I hope you found the review process to be constructive and are pleased with how the manuscript was handled editorially. We look forward to future exciting submissions from your lab.

Sincerely,
